# scRNA-seq and scATAC-seq reveal that Sertoli cell mediates spermatogenesis disorders through stage-specific communications in non-obstructive azoospermia

**Shimin Wang[1,2]\*[†], Hongxiang Wang[3†], Bicheng Jin[4], Hongli Yan[5], Qingliang Zheng[1]\*, Dong Zhao[2]\***

[1]Prenatal Diagnosis Center, The Eighth Affiliated Hospital, Sun Yat-sen University, Shenzhen, China; [2]Department of Gynaecology and Obstetrics, Ninth People's Hospital, Shanghai Jiao Tong University School of Medicine, Shanghai, China; [3]Department of Urology and Andrology, School of Medicine, Renji Hospital, Shanghai Jiao Tong University, Shanghai, China; [4]Department of Surgical Subject, Guizhou Electric Staff Hospital, Guiyang, China; [5]Reproductive Medicine Center, The Navy Medical University, Shanghai, China

**\*For correspondence:**
wangsm_2003@163.com (SW);
jackie4075@126.com (QZ);
hendryz@gmail.com (DZ)

[†]These authors contributed equally to this work

**Competing interest:** The authors declare that no competing interests exist.

## eLife Assessment

This study provides **valuable** scRNA-seq and scATAC-seq data for testicular tissues from patients with spermatogenesis disorders. By examining the transcriptomic and epigenetic changes in Sertoli cells, the authors uncovered key regulatory mechanisms underlying male infertility and identified potential therapeutic targets. While some of the cellular profiling results are **convincing**, the analyses for differential profiling of NOA cases and epigenomics data remain **incomplete**.

**Abstract** Non-obstructive azoospermia (NOA) belongs to male infertility due to spermatogenesis failure. However, evidence for cell type-specific abnormalities of spermatogenesis disorders in NOA remains lacking. We performed single-cell RNA sequencing (scRNA-seq) and single-cell assay for transposase-accessible chromatin sequencing (scATAC-seq) on testicular tissues from patients with obstructive azoospermia (OA) and NOA. HE staining confirmed the structural abnormalities of the seminiferous tubules in NOA patients. We identified 12 germ cell subtypes (spermatogonial stem cell-0 [SSC0], SSC1, SSC2, diffing-spermatogonia [Diffing-SPG], diffed-spermatogonia [Diffed-SPG], pre-leptotene [Pre-Lep], leptotene-zygotene [L-Z], pachytene [Pa], diplotene [Di], spermatids-1 [SPT1], SPT2, and SPT3) and 8 Sertoli cell subtypes (SC1-SC8). Among them, three novel Sertoli cell subtype phenotypes were identified, namely SC4/immature, SC7/mature, and SC8/further mature Sertoli cells. For each germ or Sertoli cell subtype, we identified unique new markers, among which immunofluorescence confirmed co-localization of ST3GAL4, A2M, ASB9, and TEX19 and DDX4 (classical marker of germ cell). PRAP1, BST2, and CCDC62 were co-expressed with SOX9 (classical marker of Sertoli cell) in testes tissues also confirmed by immunofluorescence. The interaction between germ cell subtypes and Sertoli cell subtypes exhibits stage-specific-matching pattern, as evidenced by SC1/2/5/7 involving in SSC0-2 development, SC3 participating in the whole process of spermiogenesis, SC4/6 participating in Diffing and Diffed-SPG development, and SC8 involving in the final stage of SPT3. This pattern of specific interactions between subtypes of germ cell and

Sertoli cell was confirmed by immunofluorescence of novel markers in testes tissues. The interaction was mainly regulated by the Notch1/2/3 signaling. Our study profiled the single-cell transcriptome of human spermatogenesis and provided many potential molecular markers for developing testicular puncture-specific marker kits for NOA patients.

## Introduction

Infertility is a common disease affecting approximately 15% of couples, and 50% of cases in infertility is attributed to male factor infertility (*Agarwal et al., 2015*). Male infertility is mainly associated with sexual dysfunction, endocrine, varicocele, and reproductive system infection (*Naz and Kamal, 2017*). Azoospermia includes obstructive azoospermia (OA) and non-obstructive azoospermia (NOA). NOA accounts for about 1% of adult males and 10–15% of infertile males (*Fakhro et al., 2018*). The etiology of NOA is varied, including cryptorchidism, post-pubertal mumps orchitis, and prior testicular torsion (*Yao et al., 2022*). At present, assisted reproductive technology, such as microdissection testicular sperm extraction and intracytoplasmic sperm injection, has made certain progress, providing help for patients to obtain biological offspring. At the same time, spermatogonial stem cell culture and transplantation technology and testicular tissue transplantation technology have also made some progress, but the clinical application is still not mature (*Gassei and Orwig, 2016*). At present, the etiology and mechanism of NOA patients are still unknown, which poses a challenge to treatment. Therefore, it is particularly urgent to further study the molecular mechanisms affecting spermatogenesis in NOA patients.

The formation of mature spermatozoa can be delineated into three stages: the proliferation of spermatogonial cells, meiotic division of spermatocytes, and the maturation of sperm cells (*Jégou, 1993*). The seminiferous tubule comprises three distinct cell types: Sertoli cells, extending from the basal membrane to the tubular lumen; germ cells of various generations; and peritubular cells, surrounding Sertoli cells and germ cells, isolated from Sertoli cells by the extracellular matrix. In most mammals, germ cells undergo continuous renewal, while Sertoli cells cease division during the pubertal developmental period, forming the seminiferous epithelium. The lining of the seminiferous epithelium is uniquely intricate within a complex tissue structure. At any given point in a seminiferous tubule, several generations of germ cells develop simultaneously in the process of contacting Sertoli cells from the basal epithelium to the apex. The evolution of each generation of germ cells is strictly synchronized with others, resulting in the formation of specific cell associations or stages at a particular segment of the tubule. The complete temporal sequence of these cell associations or stages until the reappearance of the initial association in a defined region of the tubule is referred to as the seminiferous epithelial cycle (*Hess and Moore, 1993*). Adult spermatogonial stem cells are distributed along the basal membrane at the base of the seminiferous tubule, necessitating a delicate balance between self-renewal and differentiation. Throughout this process, only Sertoli cells consistently maintain physical contact with sperm cells. Consequently, we posit that the entire process of spermatogenesis inevitably involves intense information exchange with the occurrence of Sertoli cells. However, the specifics of this interactive process remain incompletely understood at present.

Single-cell transcriptome sequencing (scRNA-seq) technology is used for high-throughput molecular detection of a single cell and exploring the whole-gene expression profiles of a single cell (*Tan et al., 2020*; *Zhao et al., 2020*). It can delineate the existence of rare cell subsets and the degree of cellular heterogeneity (*Tirumalasetty et al., 2024*; *Vértesy et al., 2018*). *Liao et al., 2019* performed scRNA-seq analysis on mouse testicular germ cells at 5.5 days after birth, revealing the heterogeneity of gene expression in undifferentiated spermatogonial cells. *Guo et al., 2018* performed scRNA-seq of testicular cells from young adults and found that there were five different spermatogonial states accompanying human spermatogonial differentiation. In addition, single-cell assay for transposase-accessible chromatin sequencing (scATAC-seq) is a widely used method to obtain a genome-wide snapshot of chromatin accessibility, signatures of active transcription and transcription factor (TF) binding (*Mimitou et al., 2021*). Cellular identity is strongly affected by the epigenetic wiring of the cell, which is measured by scATAC-seq (*Pervolarakis et al., 2020*). Therefore, scRNA-seq and scATAC-seq can be used to analyze spermatogenesis.

In this study, we performed an integrative analysis of scRNA-seq and scATAC-seq in testicular tissues from two OA and three NOA patients. Cellular heterogeneity and marker gene identification

results were analyzed, and the potential functional mechanism was inferred. Altogether, our results provide a comprehensive understanding of the maturation of the spermatogenic microenvironment and the mechanisms underlying pathogenesis, offering novel targets for NOA treatment strategies.

## Results

### scRNA-seq analysis of human testicular tissues

Testicular histology was examined by HE staining. In the OA1-P1 group, seminiferous tubules displayed normal structure with well-organized layers of germ cells at various stages of spermatogenesis, including mature spermatozoa. In the OA2-P2 group, germ cells and spermatozoa were present but show disorganized arrangement, with sloughed cells observed in the lumen. In the NOA1-P3 group, no spermatozoa were present, but a small number of spermatocytes and spermatogonia were visible, along with various stages of spermatogenesis. In the NOA2-P4 group, germ cells and spermatozoa were present but exhibit disordered arrangement, with sloughed cells in the lumen. In the NOA3-P5 group, only Sertoli cells were observed, indicating impaired spermatogenesis (*Figure 1A*). To characterize the diversity of testicular cells, we performed scRNA-seq and scATAC-seq on testicular samples from three NOA patients and two OA patients (*Figure 1B*). After removing low-quality cells, we obtained 23,889 cells for scRNA-seq (*Supplementary file 1*). Notably, the quantity of annotated germ cells in patients categorized as NOA1, NOA2, and NOA3 gradually decreased to zero (*Supplementary file 1*). This observation is in complete accordance with the clinical features exhibited by the three patients, providing evidence for the reliability of our scRNA-seq data. UMAP analysis of scRNA-seq revealed 12 major cell types, including germ cells, Leydig cells, Sertoli cells, endothelial cells, peritubular myoid cells (PMCs), smooth muscle, Schwann cell, macrophage, mast cells, T cells, B cells, and plasma cells (*Figure 1C*). A total of 41,708 high-quality cells were obtained from scATAC-seq analysis (*Supplementary file 2*), and they were also annotated with B cells, endothelial cells, PMCs, Schwann cell, smooth muscle, T cells, germ cells, Leydig cells, macrophage, and Sertoli cells, which was similar to the scRNA-seq results (*Figure 1D*). Heatmap of marker genes in scRNA-seq data revealed that cell heterogeneity was obvious (*Figure 1E*). The well-known cell-type markers were utilized to determine the cell clusters (*Figure 1F*). For instance, we observed that Sertoli cells, in addition to specifically expressing classical markers AMH and SOX9 (*Fröjdman et al., 2000*; *Lasala et al., 2011*), also exhibited strong expression of APOA1 (*Figure 1F*). Furthermore, we identified that the PMCs not only expressed the classical marker MYH11 (*Lottrup et al., 2014*), which is concurrently highly expressed in smooth muscle cells, displaying limited specificity, but also demonstrated higher specificity in the expression of DPEP1, suggesting its potential as a specific marker for PMCs. Similar to the scRNA-seq results, the activity of these markers was high in the corresponding cell subtypes in scATAC-seq data (*Figure 1—figure supplement 1A*). *Figure 1—figure supplement 1B* shows the chromatin accessibility of these markers within the cells. Integrating scATAC-seq and scRNA-seq analysis revealed substantial overlap between the cells profiled in the scRNA-seq and scATAC-seq data, implying that for most cell populations shifts in chromatin accessibility were aligned with coordinated alterations in gene expression levels (*Figure 1—figure supplement 2*). These findings indicate that the data from this study can serve for cellular classification.

### Germ cell subtypes reveal distinct molecular features

To explore the heterogeneity of germ cells during normal development, we re-clustered the germ cells and identified 12 subpopulations, including spermatogonial stem cell-0 (SSC0), SSC1, SSC2, and Diffing_SPG, Diffed_SPG, Pre-Leptotene, Leptotene_Zygotene, Pachytene, Diplotene, SPT1, SPT2, and SPT3 (*Figure 2A*). Integrating scATAC-seq and scRNA-seq analysis showed that the distributions between the same cell types were very alike (*Figure 2—figure supplement 1*). Heatmap of scRNA-seq results showed the markers successfully distinguished between different germ cell subpopulations (*Figure 2—figure supplement 2A*). Violin plots showed germ cell subtype marker gene expression in different germ cell subpopulations (*Figure 2—figure supplement 2B*). Heatmap of germ cell markers and motifs in scATAC-seq results is shown in *Figure 2—figure supplement 3A and B*. Subsequently, we attempted to characterize the sequential developmental relationships among 12 germ cell subpopulations through pseudotime analysis, approaching this issue from a developmental perspective. As shown in *Figure 2B*, germ cells could be divided into three states, which was in accordance with the

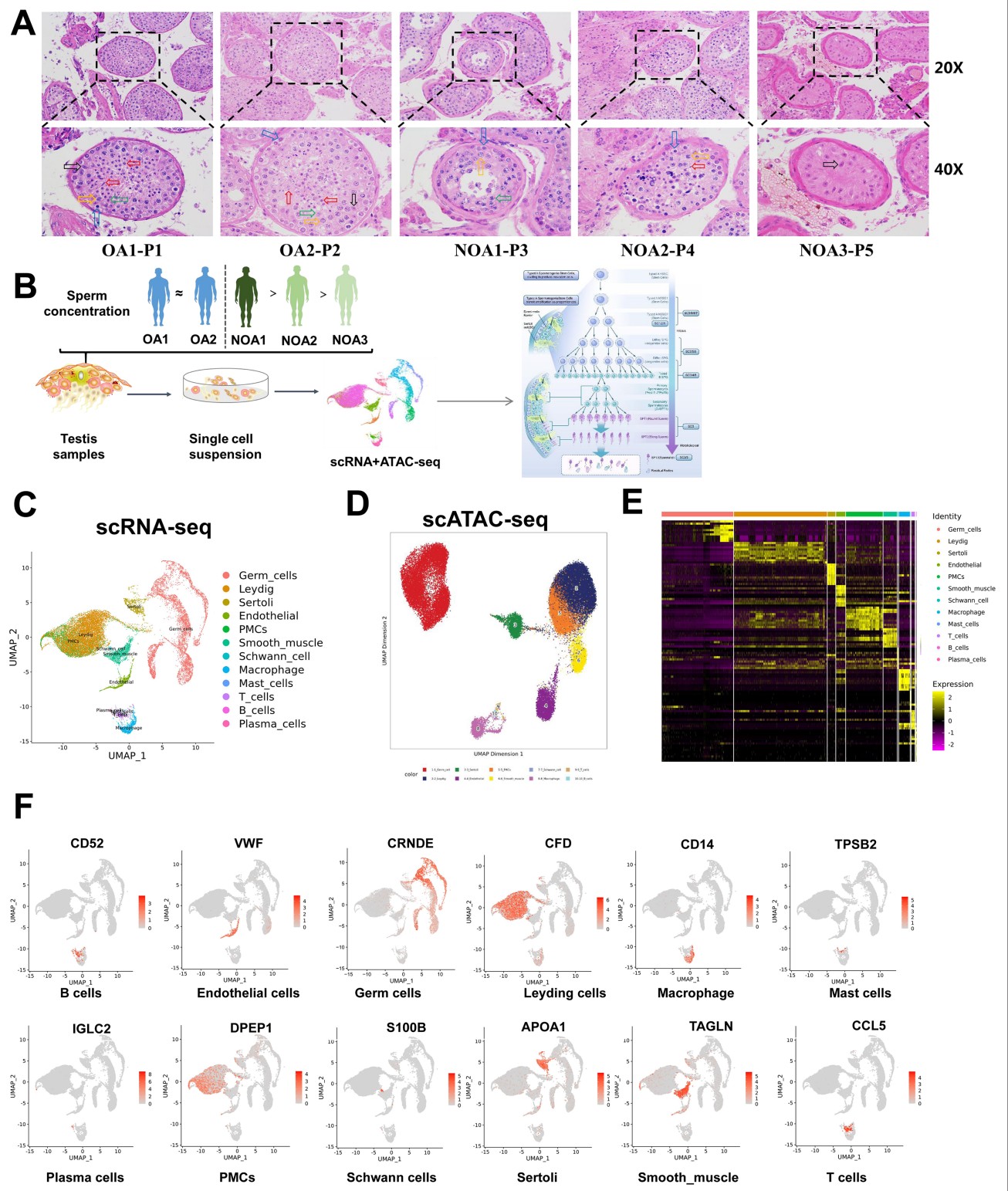

**Figure 1.** Overview of major cell types and cellular attributes of testes of obstructive azoospermia (OA) and non-obstructive azoospermia (NOA) patients. (**A**) Representative images of HE staining of testicular tissues of OA and NOA patients. (**B**) Schematic of the experimental design for scRNA-seq and scATAC-seq. (**C**) UMAP analysis of human testicular cells in scRNA-seq results. (**D**) UMAP analysis of human testicular cells in scATAC-seq results. (**E**) Heatmap of expression of markers for the 12 cell types. (**F**) UMAP analysis of testicular cell population.

The online version of this article includes the following figure supplement(s) for figure 1:

*Figure 1 continued on next page*

*Figure 1 continued*

**Figure supplement 1.** scATAC-seq analysis of the entire cell subpopulation.

**Figure supplement 2.** Integrating scRNA-seq and scATAC-seq analysis of all cell types.

biological sequence of continuous transition of spermatogenesis. States 2 and 3 primarily consist of SSC and Diffed SPG cells, and these cells are not completely segregated in terms of the cell states, particularly with SSC0-SSC2 positioned at two initial branches without complete separation (*Figure 2*, *Figure 2—figure supplement 2*). This suggests a potential cyclical and self-renewing state that is not entirely independent between SSC1 and SSC2. Cells from the Pre-L to SPT3 stages all belong to state 1, following the developmental chronological order of sperm generation. Therefore, the phenomenon of distinct subpopulations existing independently while sharing markers with other subpopulations suggests that germ cells from different subtypes may not have fully differentiated into mature states during the course of evolution, remaining in an intermediate state of continuous differentiation.

Furthermore, we presented the expression distribution of markers for different subpopulations across three developmental stages in the chronological order of germ cell maturation (*Figure 2C*). Interestingly, the positions of different subpopulations of markers on UMAP show a trend of right-to-left panning as the arrows point to the temporal developmental order of sperm cells, which was supported by RNA velocity stream UMAP (*Figure 2D*). Among them, SPT1 cells can be clearly distin-guished from SPT2-3 cells using FAM24A, but between SPT2 and SPT3 cells, there is no precise and specific marker that completely segregates these two cell types. We speculate that this lack of differ-entiation may be due to the intense morphological changes occurring in the sperm cells during this period, resulting in relatively minor differences in gene expression.

Finally, to further validate the reliability of the identified new potential markers of germ cells, we co-stained them with a classical marker DDX4 using immunofluorescence. We discovered that ST3GAL4, A2M, and DDX4 were co-expressed in testes tissues. ASB9 and TEX19 were also co-ex-pressed with DDX4 in testes tissues. TSSK6 exhibited a similar expression pattern to that of DDX4 in testes tissues (*Figure 2E*). In addition, closely resembling their expression patterns in different subpopulations, we found that the region transcription start/promoter site to A2M and ASB9 was accessible in the SSC0, SSC2, and Diffing_SPG but was not accessible in other cell subtypes, with much higher levels of openness (*Figure 2F*). The region transcription start site to TEX19 was acces-sible in the Diffed_SPG but was not accessible in other cell subtypes, which also closely resembled the TEX19 expression pattern. Collectively, these results suggest that these new markers localize to testicular tissue and have the same ability to label sperm cells as DDX4.

## Transcriptome differences in germ cells between the OA and NOA groups

To gain a deeper understanding of the role of germ cells in spermatogenesis, we investigated the function of differentially expressed genes (DEGs) in germ cells between the NOA and OA groups. The volcano map showed that there were 665 DEGs between the OA and NOA groups, including 367 downregulated genes and 298 upregulated genes (*Figure 2—figure supplement 4A*). The top 5 DEGs are illustrated using a dot-plot (*Figure 2—figure supplement 4B*). GO analysis showed that DEGs were mainly associated with meiotic cell cycle and meiotic cell cycle process (*Figure 2—figure supplement 4C*). KEGG analysis showed that DEGs were primarily involved in cell cycle and DNA replication (*Figure 2—figure supplement 4D*). The differential peaks in germ cells between the OA and NOA groups can be observed in the circos plot (*Figure 2—figure supplement 4E*). Subsequently, we further analyzed the peak changes of the top 5 DEGs involved in the cell cycle in germ cells between the two groups. The results showed that the peak signal intensity of CDC6 was increased in the OA group compared with the NOA group (*Figure 2—figure supplement 5A–E*). Overall, CDC6 may play an important role in spermatogenesis.

## Cell cycle and transcriptional dynamics analysis of germ cell differentiation

Most adult tissues are sustained by resident adult stem cells responsible for maintaining tissue function and integrity. As aging or damaged cells undergo apoptosis, resident adult stem cells are activated

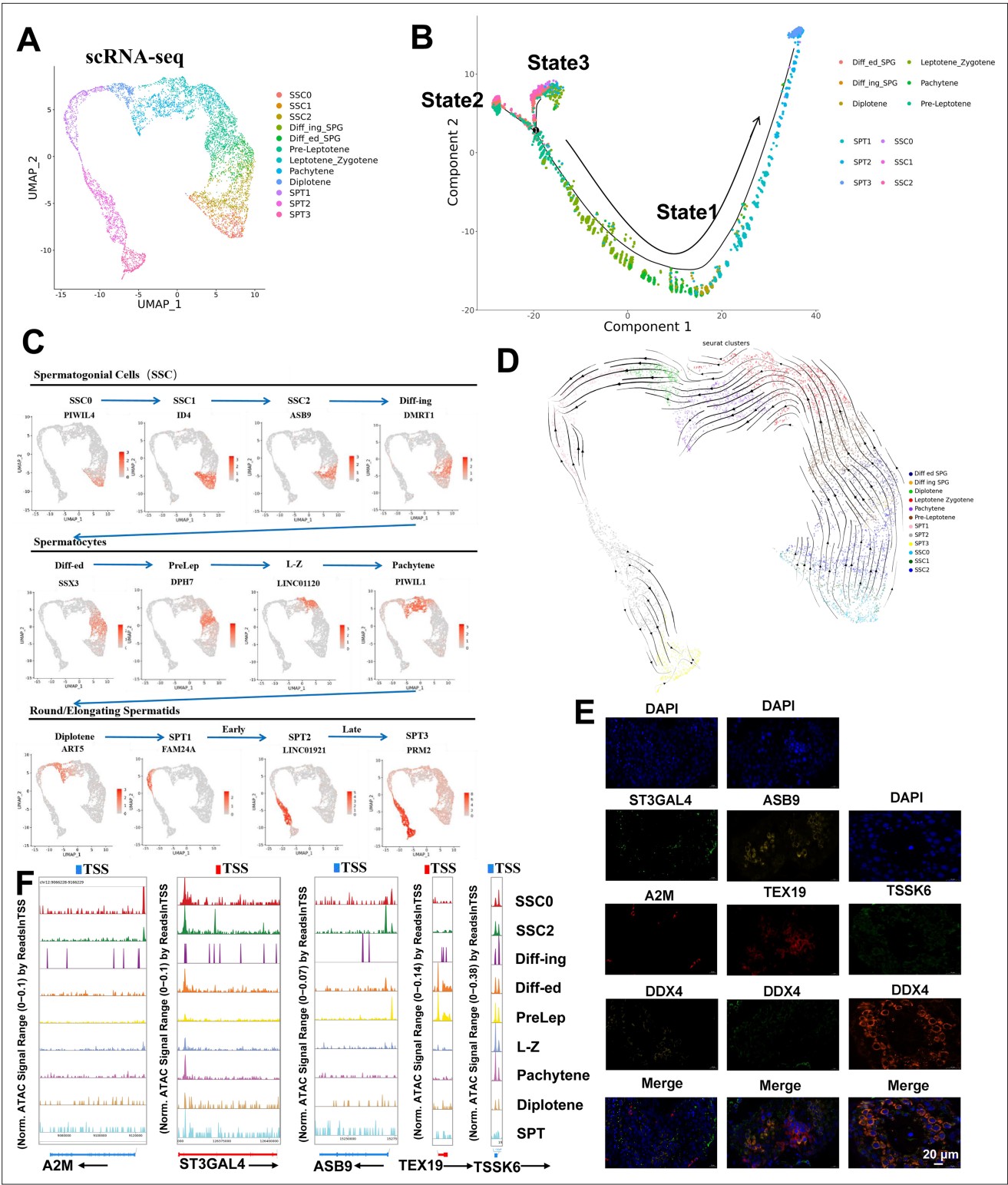

**Figure 2.** Identification of germ cell subtypes. (**A**) UMAP analysis of 12 germ cell subsets in scRNA-seq results. (**B**) Pseudotime analysis of 12 germ cell subpopulations. (**C**) Gene expression patterns of marker genes on UMAP plots. (**D**) State of germ cell subsets using pseudotime analysis. (**E**) Immunofluorescence double staining of markers of newly identified spermatocyte subpopulations with the classical marker DDX4. (**F**) Accessibility peak plots of markers for newly identified germ cell subpopulations.

The online version of this article includes the following figure supplement(s) for figure 2:

**Figure supplement 1.** Integrating scRNA-seq and scATAC-seq analysis of germ cell subtypes.

*Figure 2 continued on next page*

*Figure 2 continued*

**Figure supplement 2.** Analysis of germ cell markers in spermatogenesis.

**Figure supplement 3.** scATAC-seq clustering.

**Figure supplement 4.** Distribution characteristics of differentially expressed genes (DEGs) in germ cells from the obstructive azoospermia (OA) and non-obstructive azoospermia (NOA) groups.

**Figure supplement 5.** Normalized chromatin accessibility of genes involved in cell cycle in germ cells between the obstructive azoospermia (OA) and non-obstructive azoospermia (NOA) groups.

to generate new cells. Within various tissues, a dichotomy exists between rapidly cycling (involved in tissue repair, EOMES$^-$GFRA1$^+$) and quiescent stem cells (reserve cells, EOMES$^+$GFRA1$^+$) (*Sharma et al., 2018*). To investigate the cellular developmental states of distinct sperm subpopulations, we conducted an analysis of the cell cycle in sperm cells. As shown in *Figure 3A and B*, SSC0/1 and SSC2 are quiescent stem cells and rapidly cycling cells, respectively. Furthermore, to explore the biological function of germ cells, we used transcriptional dynamics to analyze the important events and time points in the whole transcriptional process of spermatogenesis. Maintenance of SSC pluripotency, whether in the G0/G1 phase and G2/M arrest SSC0/SSC1, or rapid cycling SSC2, is transcriptionally governed by the downregulation of ID4 (*Figure 3C*). The pivotal driving genes promoting the transition of SSC2 into differentiating Diffing SPG cells and progressing through mitosis are upregulated DMRT1 and PABPC4 (*Figure 3C*). As BEND2 takes the central regulatory position, germ cell development officially proceeds into meiosis, marking the transitioned cells (Diffed SPG) in a mixed transitional state concurrently undergoing both mitosis and meiosis (*Figure 3C*). The molecular markers distinguishing between the PreLep and SPT3 subpopulations included C18orf63, MNS1, CCDC42, LDHC, TMIGD3, AC113189.2, and FLJ40194 (*Figure 3C*). Results showed that there is no distinctly discernible regulatory factor between PreLep and L-Z, with C18orf63 being a crucial differentiator, upregulated in PreLep and downregulated in L-Z. Finally, we summarized the spermatogenesis at each stage and major events in the cell cycle (*Figure 3D*).

Subsequently, beam analysis was performed to unveil the critical transcription factors involved in sperm differentiation. *Figure 3—figure supplement 1A* hierarchically clustered the top 50 transcription factors into three groups, presenting an expression heatmap of fate-determining genes associated with the differentiation from SSC0 to SPG state. The expression pattern of transcription factors within cluster 1 gradually increased along the temporal trajectory (*Figure 3—figure supplement 1B*), predominantly involved in cytoplasmic translation GO functions (*Figure 3—figure supplement 1C*); cluster 2 exhibited an initial increase followed by a decline (*Figure 3—figure supplement 1B*), governing positive regulation of translation and regulation of G2M transition of mitotic cell cycle (*Figure 3—figure supplement 1C*); and transcription factors within cluster 3 showed a sustained decrease in expression (*Figure 3—figure supplement 1B*) and were enriched in cell cycle and meiotic cell cycle-related GO terms (*Figure 3—figure supplement 1C*). As shown in *Figure 3E*, PIWIL4, C19orf84, and ST3GAL4 were specifically expressed in SSC0 (belong to cluster 2). ID4, TCF3, and UTF1 were specifically expressed in SSC1 (belong to cluster 2). DMRT1, ASB9, and NANOS3 were the main specifically expressed genes in SSC2 (belong to cluster 3). DYNLL1, APRT, and MKI67 were specifically expressed in Diffing-SPG (belong to cluster 1/2/3). CTCFL, TEX19, and SMC1B were specifically expressed in Diffed-SPG (belong to cluster 1). Heatmaps of transcription factors showed that the transcription of SSC1 and SSC2 was very active during reproductive development (*Figure 3—figure supplement 1D*). In contrast, SPT2/3 was almost transcribed in silence. Diffing/ed served as a connecting link between the preceding and the following, but its transcriptional state was closer to SSC (*Figure 3—figure supplement 1D*). In conclusion, we obtained 50 transcription factors that are closely related to sperm differentiation.

## Sertoli cells from human testicular tissues are composed of heterogeneity population

To explore the heterogeneity of Sertoli cells, we reclustered the Sertoli cells and identified eight subclusters (*Figure 4A*). Integrating scATAC-seq and scRNA-seq analysis, the distributions between the same cell types were very alike (*Figure 4—figure supplement 1*). The Sertoli cell marker genes were utilized to determine the Sertoli cell clusters using heatmap (*Figure 4B*) and violin plots (*Figure 4C*). Of note,

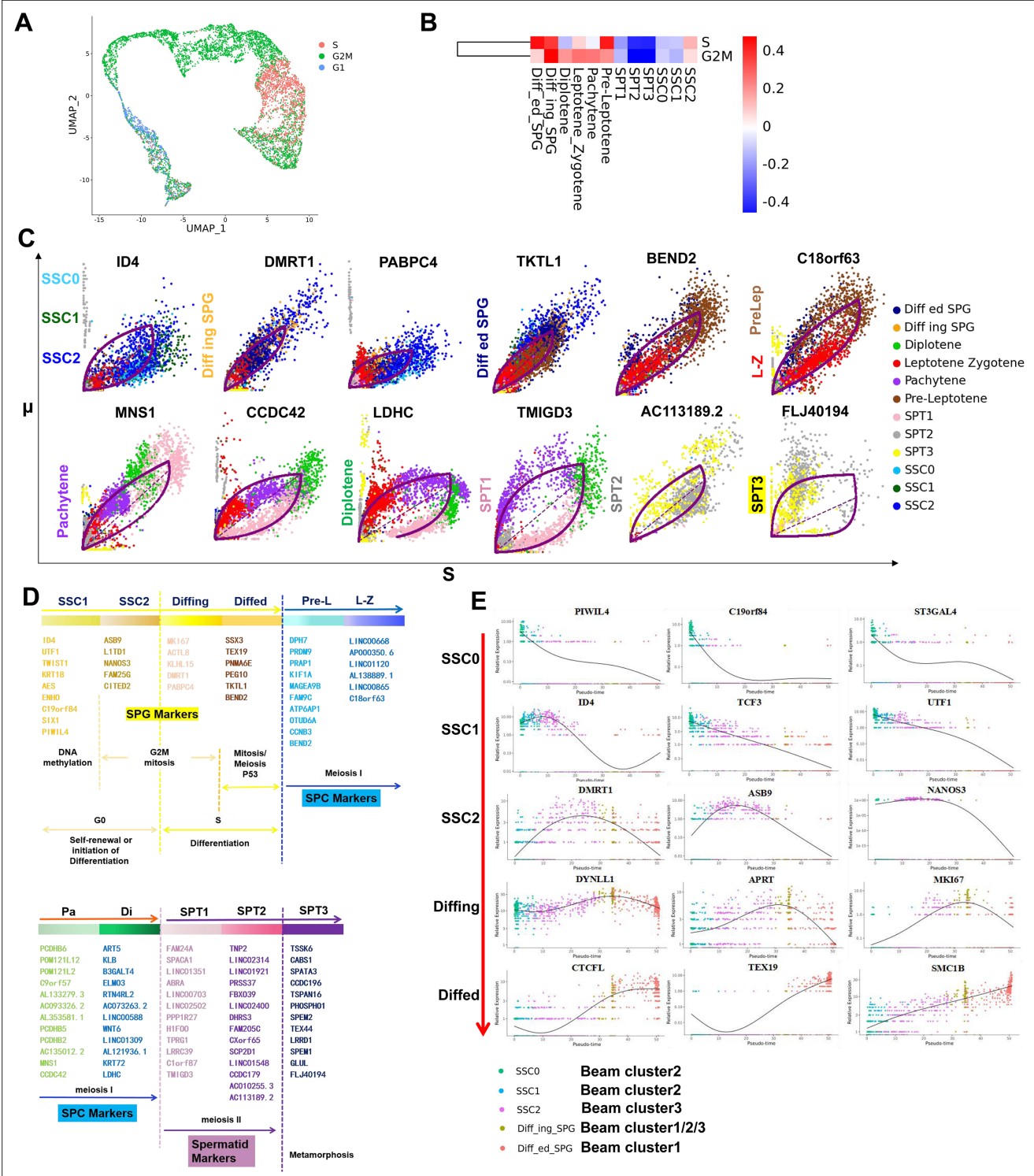

**Figure 3.** Analysis of germ cells at different stages. (**A**) Germ cell subsets at different stages of the cell cycle. (**B**) Heatmap of cell cycle in germ cells. (**C**) Transcriptional dynamics of driver genes of germ cell subtypes. (**D**) The main stage of spermatogenesis. (**E**) Beam analysis of differentially expressed genes at different stages of spermatogonial stem cell (SSC).

The online version of this article includes the following figure supplement(s) for figure 3:

**Figure supplement 1.** Germ cell function analysis and epigenetic analysis.

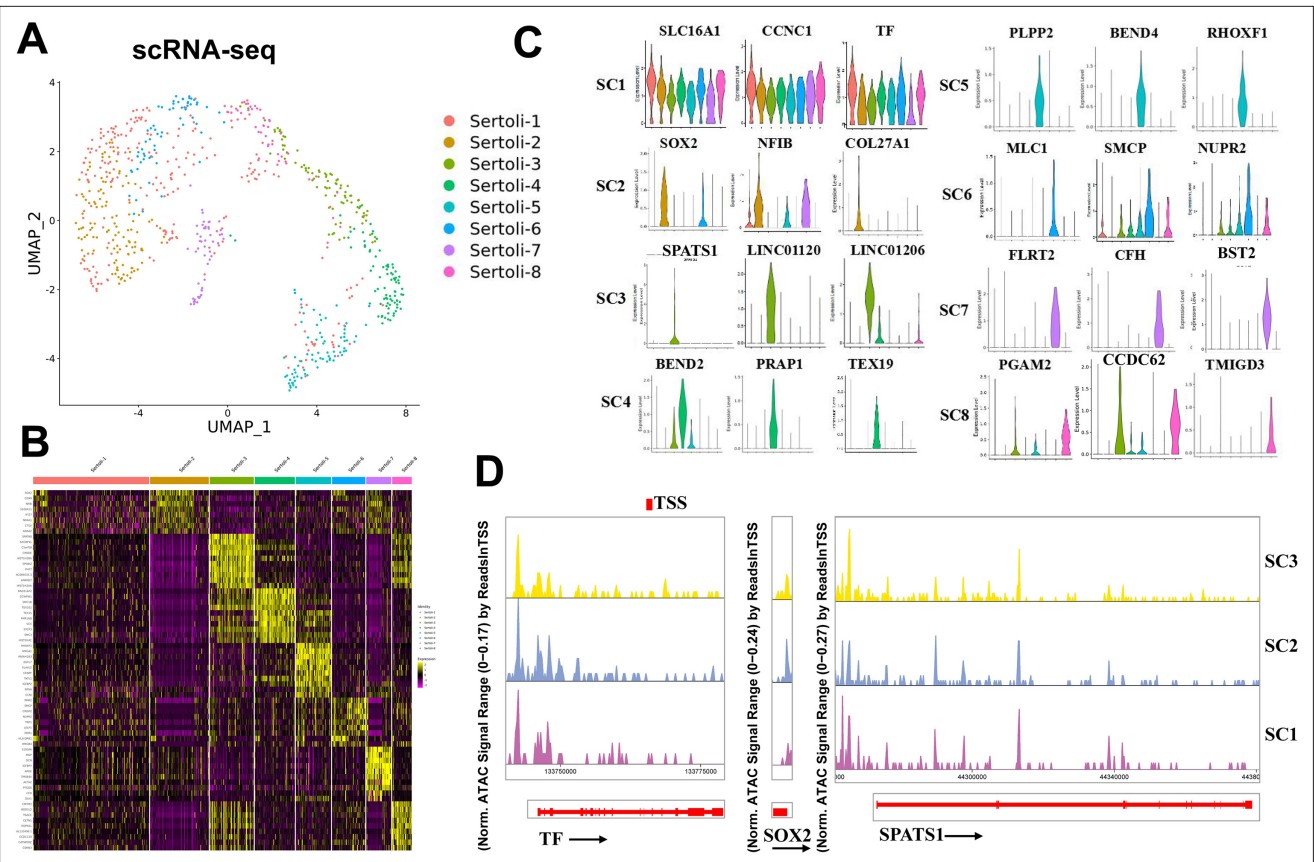

**Figure 4.** Identification of Sertoli cell subtypes and epigenetic analysis. (**A**) UMAP analysis showed eight Sertoli cell subsets in scRNA-seq results. (**B**) Heatmap of Sertoli cell subtype marker genes in eight Sertoli cell subsets. (**C**) Gene expression patterns of Sertoli cell subtype marker genes on violin plots. (**D**) Accessibility peak plots of markers for newly identified Sertoli cell subpopulations.

The online version of this article includes the following figure supplement(s) for figure 4:

**Figure supplement 1.** Integrating scRNA-seq and scATAC-seq analysis of Sertoli cell subtypes.

**Figure supplement 2.** Distribution characteristics of differentially expressed genes (DEGs) in Sertoli cells from the obstructive azoospermia (OA) and non-obstructive azoospermia (NOA) groups.

**Figure supplement 3.** Normalized chromatin accessibility of genes involved in cell cycle in Sertoli cells between obstructive azoospermia (OA) and non-obstructive azoospermia (NOA) groups.

no cluster-specific distinctive features were identified in SC1 as the marker genes (TF, SLC16A1, and CCNL1) of SC1 were also highly expressed in other Sertoli cell subclusters (*Figure 4C*). SOX2, NF1B, and COL27A1 were preferentially expressed in SC2. SPATS1, LINC01120, and LINC01206 were specifically expressed in SC3. BEND2, PRAP1, and TEX1 were specifically expressed in SC4. PLPP2, BEND4, and RHOXF1 were specifically expressed in SC5. MLC1, SMCP, and NUPR2 were highly expressed in SC6. FLRT2, CFH, and BST2 were specifically expressed in SC7. TMIGD3, PGAM2, and CCDC62 were specifically expressed in SC8. Subsequently, to delve deeper into the role of Sertoli cells in spermatogenesis, we conducted a comprehensive analysis of DEGs in Sertoli cells between the NOA and OA groups. The volcano plot revealed a total of 414 DEGs, comprising 291 downregulated and 123 upregulated genes in the OA group (*Figure 4—figure supplement 2A*). A dotplot was employed to illustrate the top 5 DEGs between the NOA and OA groups (*Figure 4—figure supplement 2B*). GO analysis indicated that these DEGs were predominantly associated with meiotic cell cycle, meiotic cell cycle process, and germ cell development, as depicted in *Figure 4—figure supplement 2C*. Additionally, KEGG analysis revealed that the DEGs were primarily involved in cell cycle, cytokine–cytokine receptor interaction, and TNF signaling pathway, as shown in *Figure 4—figure supplement 2D*. The heatmap revealed that there were 13 differential peaks in Sertoli cells between the OA and NOA groups (*Figure 4—figure supplement 2E*). We then further analyzed the peak changes of the top 5

DEGs involved in the cell cycle in Sertoli cells between the two groups. The results uncovered that the peak signal intensity of BUB18 and SMC1B was increased in the OA group compared with the NOA group (*Figure 4—figure supplement 3A–E*).

Cell differentiation is accompanied by the expression of genes controlled by *cis*-regulatory elements, which must be in an accessible state to function properly. Therefore, scATAC-seq analysis was performed on the same Sertoli cells as those used in scRNA-seq analysis, and a chromatin accessibility landscape for individual cell marker was delineated (*Figure 4D*). Because the number of cells in the SC4-SC8 subpopulation was too small to calculate a peak plot, only the peak plots of markers in the SC1-SC3 subpopulations are displayed. The promoter regions of TF were all highly accessible in the SC1-SC3 subpopulations, and relatively higher openness was observed for SOX2 in the SC2 subgroup. In addition to the high openness we observed in the promoter region of the SPATS1, there were also multiple accessible regions within the gene in the SC1-SC3 subpopulations.

## Developmental trajectories of Sertoli cells and identified three novel subpopulations

To analyze the origin and maturation process of Sertoli cells, pseudotime trajectory analysis was carried out. The trajectory was determined to initiate with state 3 as beginning and reached a terminally differentiate state of state 2 and state 1 (*Figure 5A*). According to *Figure 5B*, state 1 mainly included SC3/4/5 clusters, state 2 mainly included SC2/6/8 clusters, and state 3 mainly included SC1/7 clusters. *Figure 5C* demonstrates RNA velocity heatmap of dynamic evolutionary Sertoli cells in the three states. Although most studies have categorized the development of Sertoli cells after birth into two stages, immature and mature, the existence of an intermediate or transitional state remains largely unknown. However, in this study, we identified three distinct states of Sertoli cells, with state 1 expressing markers EGR3, CTSL, PCNA, MK167, and KRT18 associated with immature Sertoli cells (*Zhao et al., 2020*), while state 3 expressed markers, such as TF, HOPX, and DEF119, indicative of mature Sertoli cells (*Zhao et al., 2020*). Consequently, we defined state 1 and state 3 as immature and mature Sertoli cells, respectively. State 2, falling between immature and mature states, does not belong to either category. Therefore, it is defined as a further maturation Sertoli cells, aligning with previously reported findings in the literature (*Guo et al., 2021*; *Zhao et al., 2020*). According to the latent time of RNA rate, the differentiation process is SC3, SC4, SC5, SC2, SC6, SC8, SC1, and SC7 (*Figure 5D*). Based on the above result, we identified three novel Sertoli cell subtypes (namely SC4/7/8) and their specific markers, and defined SC4 (PRAP1) as immature, SC7 (BST2) as mature, and SC8 (CCDC62) as further mature (*Figure 5D*). To validate these novel identified markers in Sertoli cells, we co-stained them with SOX9, a classical marker of Sertoli cells. As shown in *Figure 5E*, we discovered that PRAP1, BST2, and CCDC62 were co-expressed with SOX9 in testes tissues.

GO analysis of Sertoli cells in different states is shown in *Figure 5—figure supplement 1A*. Sertoli cells in state 1 were mainly involved in cell cycle, meiotic cell cycle, and spermatogenesis. Sertoli cells in state 2 were mainly involved in platelet degranulation, lipid metabolic process, and steroid biosynthetic process. Sertoli cells in state 3 were mainly involved in extracellular matrix organization, SRP-dependent cotranslational protein targeting to membrane and viral transcription. Further, Qusage analysis was performed for Sertoli cells (*Figure 5—figure supplement 1B*). Sertoli cells in state 1 were mainly involved in cell cycle, DNA replication, and mismatch repair. Sertoli cells in state 2 were involved in D-glutamine and D-glutamate metabolism, ribosome, and valine, leucine, and isoleucine biosynthesis. Sertoli cells in state 3 participated in ECM−receptor interaction, and protein digestion and absorption. Overall, Sertoli cells performed different functions at different stages.

## Existence of Sertoli cell subtypes is more crucial for spermatogenesis than its quantity

The highly interdependent structural relationship between Sertoli cells and germ cells has long been considered evidence of their close functional association (*Griswold, 1995*). For instance, some germ cells, due to their structural affinity with Sertoli cells, cannot be fully separated, and when cultured independently in vitro, these germ cells exhibit very short survival times; however, the addition of Sertoli cells or the use of Sertoli cells-conditioned medium significantly improves the survival of germ cells (*La et al., 2018*; *Mohammadi-Sardoo et al., 2021*; *Risley and Morse-Gaudio, 1992*). To determine which subtype of Sertoli cells is more closely involved in spermatogenesis, we analyzed

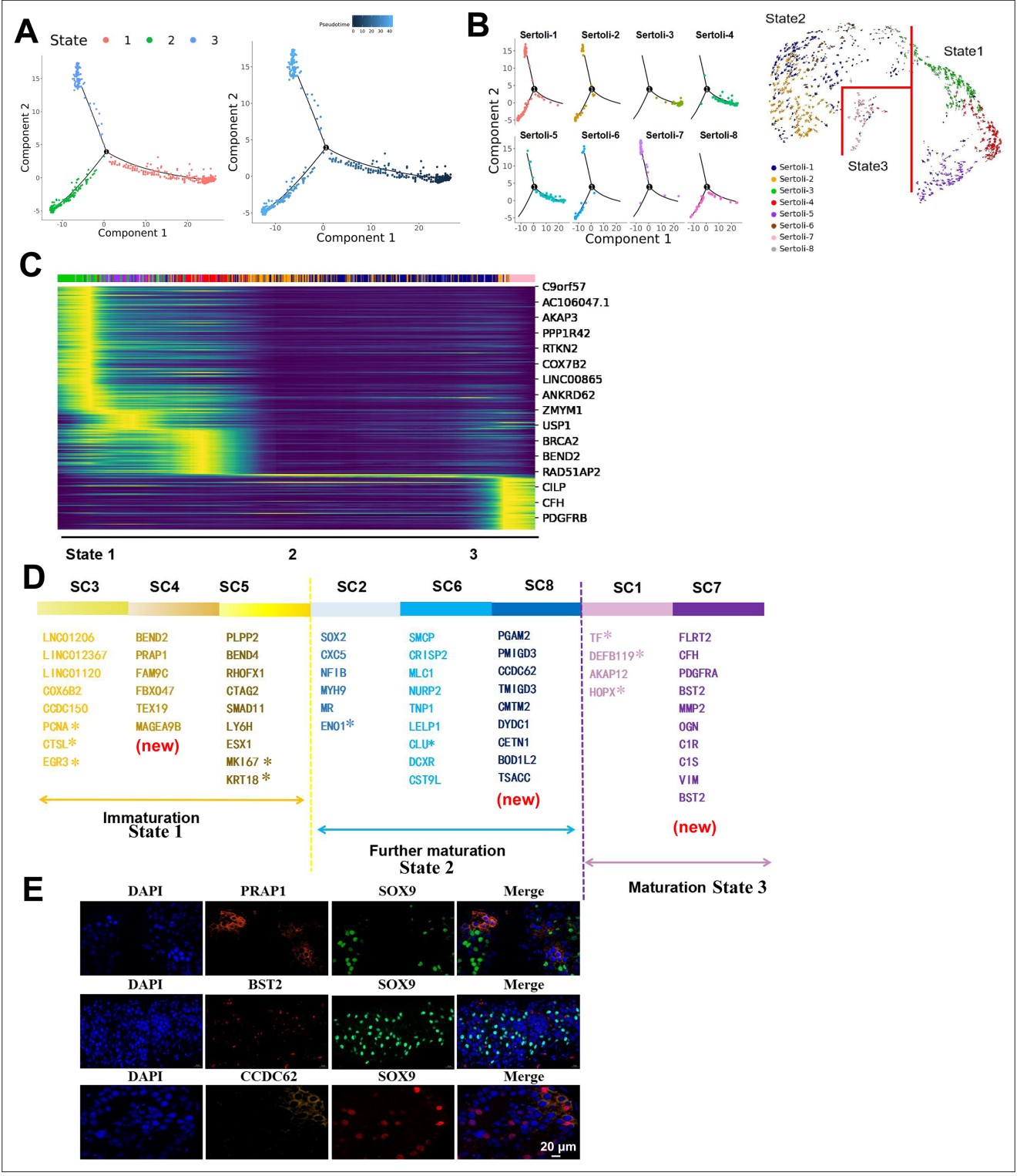

**Figure 5.** The development of Sertoli cells. (**A**) State of Sertoli cells using pseudotime analysis. (**B**) Distribution of Sertoli cell subtypes in different states. (**C**) Potential temporal differentiation of SC subgroups using RNA velocity. (**D**) Sertoli cells division diagram and representative markers. (**E**) Immunofluorescence of Sertoli cell subtype markers in testicular tissues from obstructive azoospermia (OA) patients. The scale bar represents 20 μm.

The online version of this article includes the following figure supplement(s) for figure 5:

**Figure supplement 1.** GO and Qusage analysis of different states of Sertoli cells.

spermatogenesis and Sertoli cells distribution in the NOA (experimental group) and OA (control group) groups (*Figure 6*). The number proportion of germ cell subsets in NOA1/2 and OA1/2 is shown in *Figure 6A*. The number of germ cells in NOA3 patients was 0. The proportion of Sertoli cell subsets in NOA1/2/3 and OA1/2 is shown in *Figure 6B* (*Supplementary file 3*, *Supplementary file 4*). It can be seen that the number of Sertoli cells in three NOA samples was lower than that in OA samples, indicating that the number of Sertoli cells is somewhat correlated with spermatogenesis. Interestingly, we observed a complete absence of immature Sertoli cells, especially SC3, in the testicular tissue of patients with NOA3 who exhibited a total absence of sperm, with only a small population of mature SC7 cells present (*Figure 6C and D*), suggesting that the absence of sperm in NOA3 patients may be associated with Sertoli cells SC3. For NOA2 samples, although the number of Sertoli cells was less than that of NOA3, SC3 was not missing in NOA2, so spermatogenesis was only partially affected (*Figure 6C–F*). To further elucidate the role of SC3 in spermatogenesis, we analyzed DEGs in SC3 between the NOA and OA groups. Our results identified a total of eight DEGs, with seven being downregulated and one upregulated in the OA group (*Figure 6—figure supplement 1A*). A dotplot was utilized to highlight the expression of DEGs in SC3 between the two groups (*Figure 6—figure supplement 1B*). Interestingly, compared with the NOA group, HIST1H4C was upregulated, while TMEM158, TUBA3E, and VTN were downregulated in the OA group. GO analysis revealed that these DEGs were predominantly associated with positive regulation of monocyte differentiation and negative regulation of fibrinolysis, as illustrated in *Figure 6—figure supplement 1C*. KEGG analysis demonstrated that the DEGs were primarily involved in phagosome and ECM−receptor interaction (*Figure 6—figure supplement 1D*). The heatmap revealed that there were three differential peaks in SC3 between the OA and NOA groups (*Figure 6—figure supplement 1E*). We then further analyzed the peak changes of DEGs involved in the ECM−receptor interaction in SC3 between the two groups. The results showed that the peak signal intensity of VTN was decreased in the OA group relative to the NOA group (*Figure 6—figure supplement 1F*). In conclusion, these data suggested that whether or not the critical subtype SC3 is missing might be more important for spermatogenesis than the number of Sertoli cells.

## Co-localization of subpopulations of Sertoli cells and germ cells

To explore the role of Sertoli cells in spermatogenesis, we applied CellPhoneDB to infer cellular interactions according to ligand–receptor signaling database. As shown in *Figure 6—figure supplement 2*, compared with other cell types, germ cells mainly interacted with Sertoli cells. We further performed Spearman correlation analysis to determine the relationship between Sertoli cells and germ cells. As shown in *Figure 6G*, state 1 SC3/4/5 tended to be associated with PreLep, SSC0/1/2, and Diffing and Diffed-SPG sperm cells ($R>0.72$). Interestingly, SC3 was significantly positively correlated with all germ cell subpopulations ($R>0.5$). The single-cell analysis of GSE149512 further confirmed a positive correlation between SC3 and germ cells (*Figure 6H*). These results suggested SC3 might be involved in spermatogenesis. Subsequently, to understand the association between the functions of Sertoli cells and germ cells, GO term enrichment analysis of germ cells and Sertoli cells was carried out (*Figure 6—figure supplements 3 and 4*). We found that the functions could be divided into eight categories, namely, material energy metabolism, cell cycle activity, the final stage of sperm cell formation, chemical reaction, signal communication, cell adhesion and migration, stem cells and sex differentiation activity, and stress reaction. These different events were labeled with different colors in order to quickly capture the important events occurring in the cells at each stage. As shown in *Figure 6—figure supplement 3*, we discovered that SSC0/1/2 was involved in an SRP-dependent cotranslational protein targeting to membrane and cytoplasmic translation; Diffing SPG was involved in cell division and cell cycle; Diffied SPG was involved in cell cycle and RNA splicing; Pre-Leptotene was involved in cell cycle and meiotic cell cycle; Leptotene_Zygotene was involved in cell cycle and meiotic cell cycle; Pachytene was involved in cilium assembly and spermatogenesis; Diplotene was involved in spermatogenesis and cilium assembly; SPT1 was involved in cilium assembly and flagellated sperm motility; SPT2 was involved in spermatid development and flagellated sperm motility; and SPT3 was involved in spermatid development and spermatogenesis. As shown in *Figure 6—figure supplement 4*, SC1 were mainly involved in cell differentiation, cell adhesion, and cell communication; SC2 were involved in cell migration and cell adhesion; SC3 were involved in spermatogenesis and meiotic cell cycle; SC4 were involved in meiotic cell cycle

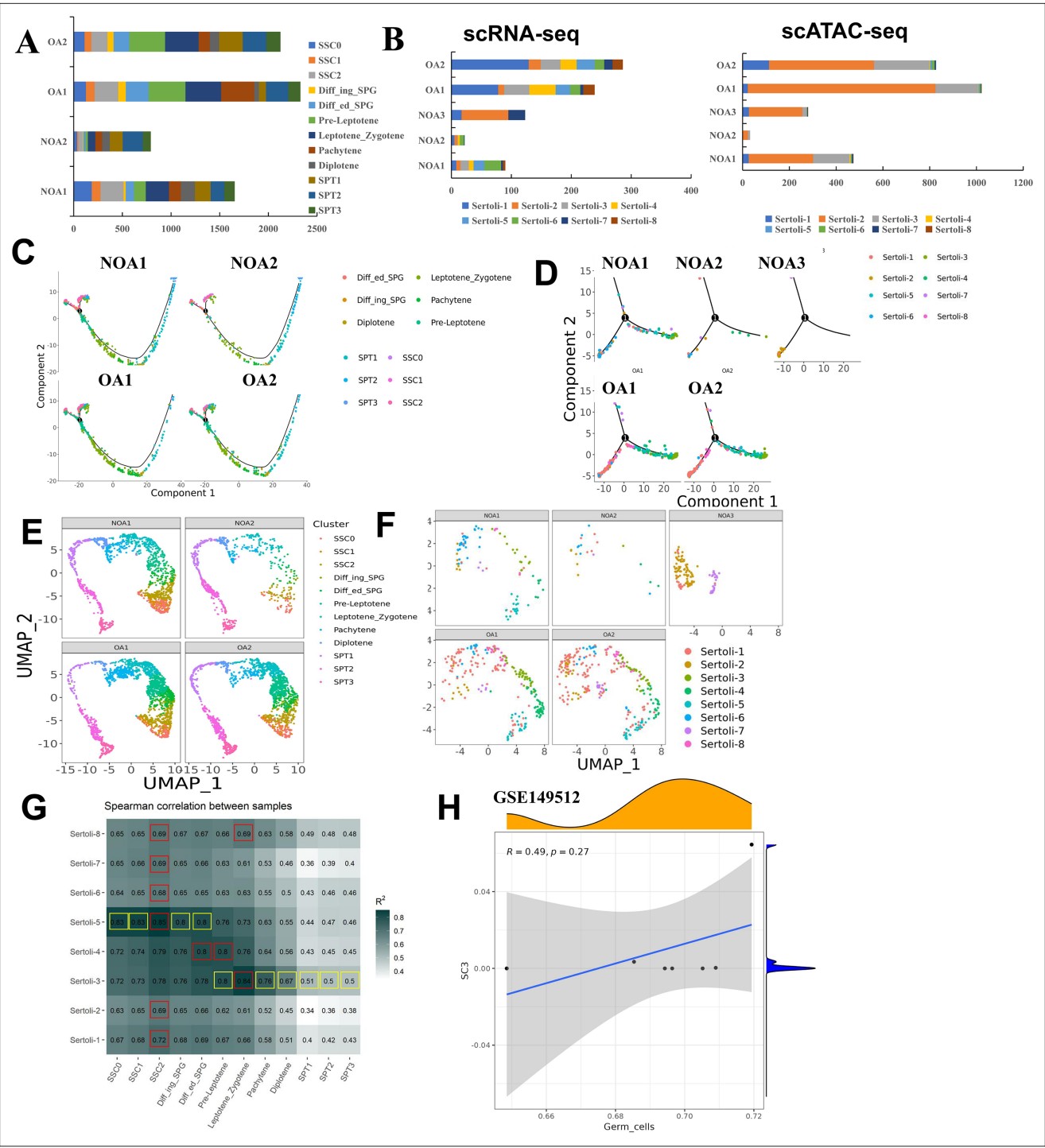

**Figure 6.** Correlation analysis of germ and Sertoli cell expression in different groups. (**A**) Cell proportion of germ cell subtypes in different groups. (**B**) Cell proportion of Sertoli cell subtypes in different groups. (C) Pseudotime analysis of germ cell subtypes in different groups. (**D**) Pseudotime analysis of Sertoli cell subtypes in different groups. (**E**) Distribution map of germ cell subtypes in different samples. (**F**) Distribution map of Sertoli cell subtypes in different samples. (**G**) Spearman correlation between germ and Sertoli cells. (**H**) GSE149512 was used to verify the correlation between SC3 and germ cells.

The online version of this article includes the following figure supplement(s) for figure 6:

**Figure supplement 1.** Distribution characteristics of differentially expressed genes (DEGs) in Sertoli cell subtype SC3 from the obstructive azoospermia (OA) and non-obstructive azoospermia (NOA) groups.

**Figure supplement 2.** The interaction between germ cells and other cells.

*Figure 6 continued on next page*

**Figure supplement 3.** GO analysis of germ cells.

**Figure supplement 4.** GO analysis of Sertoli cells.

**Figure supplement 5.** Correlation analysis of germ and Sertoli cell expression.

and positive regulation of stem cell proliferation; SC5 were involved in cell cycle and cell division; SC6 were involved in obsolete oxidation–reduction process and glutathione derivative biosynthetic process; SC7 were involved in viral transcription and translational initiation; and SC8 were involved in spermatogenesis and sperm capacitation. The above analysis indicated that germ cells were primarily involved in biological processes related to sperm formation, and Sertoli cells were mainly involved in processes related to the formation of sperm cells, suggesting a functional correlation between Sertoli cells and germ cells.

To further verify that Sertoli cell subtypes have 'stage specificity' for each stage of sperm development, we first performed HE staining using testicular tissues from the OA3-P6, NOA4-P7, and NOA5-P8 samples. The results showed that the OA3-P6 group showed some sperm, with reduced spermatogenesis, thickened basement membranes, and a high number of Sertoli cells without spermatogenic cells. The NOA4-P7 group had no sperm initially, but a few malformed sperm were observed after sampling, leading to the removal of affected seminiferous tubules. The NOA5-P8 group showed no sperm in situ (*Figure 7A*). Immunofluorescence staining in *Figure 7B* was performed using these tissues for validation. ASB9 (SSC2) was primarily expressed in a wreath-like pattern around the basement membrane of testicular tissue, particularly in the OA group, while ASB9 was barely detectable in the NOA group. SOX2 (SC2) was scattered around SSC2 (ASB9), with nuclear staining, while TF (SC1) expression was not prominent. In NOA patients, SPATS1 (SC3) expression was significantly reduced. C9orf57 (Pa) showed nuclear expression in testicular tissues, primarily extending along the basement membrane toward the spermatogenic center, and was positioned closer to the center than DDX4, suggesting its involvement in germ cell development or differentiation. BEND4, identified as a marker for SC5, showed a developmental trajectory from the basement membrane toward the spermatogenic center. ST3GAL4 was expressed in the nucleus, forming a circular pattern around the basement membrane, similar to A2M (SSC1), though A2M was more concentrated around the outer edge of the basement membrane, creating a more distinct wreath-like arrangement. In case of impaired spermatogenesis, this arrangement becomes disorganized and loses its original structure. SMCP (SC6) was concentrated in the midpiece region of the bright blue sperm cell tail. In the OA group, SSC1 (A2M) was sparsely arranged in a rosette pattern around the basement membrane, but in the NOA group, it appeared more scattered. SSC2 (ASB9) expression was not prominent. BST2 (SC7) was a transmembrane protein primarily localized on the cell membrane. In the OA group, A2M (SSC1) was distinctly arranged in a wreath-like pattern around the basement membrane, with expression levels significantly higher than ASB9 (SSC2). TSSK6 (SPT3) was primarily expressed in OA3-P6, while CCDC62 (SC8) was more abundantly expressed in NOA4-P7, with ASB9 (SCC2) showing minimal expression. Taken together, germ cells of a particular stage tended to co-localize with Sertoli cells of the corresponding stages. Germ cells and Sertoli cells at each differentiation stage were functionally heterogeneous and stage-specific (*Figure 8*). We speculate that each stage of sperm development requires the assistance of Sertoli cells to complete the corresponding stage of sperm development.

## Notch1/2/3 signaling participates in the interaction between Sertoli cells and germ cells

To explore the mechanism of interaction between Sertoli cells and germ cells, we conducted Cell-Phone analysis. Results showed that Notch1/2/3 signaling was involved in the interaction between Sertoli cells and germ cells (*Figure 6—figure supplement 5A*). Interestingly, a chromatin accessibility landscape in scATAC-seq data showed that promoter regions of Notch1 were only accessible in the SC1-SC3 subpopulations, not in the germ cells (*Figure 6—figure supplement 5B*). The openness of the promoter region of Notch2 was observed not only in Sertoli cells but also in SSC2 and Pachytene. Notch3 accessibility was only observed within the gene in different cell subtypes. Overall, Notch1/2/3 signaling might be involved in the interaction between Sertoli cells and germ cells.

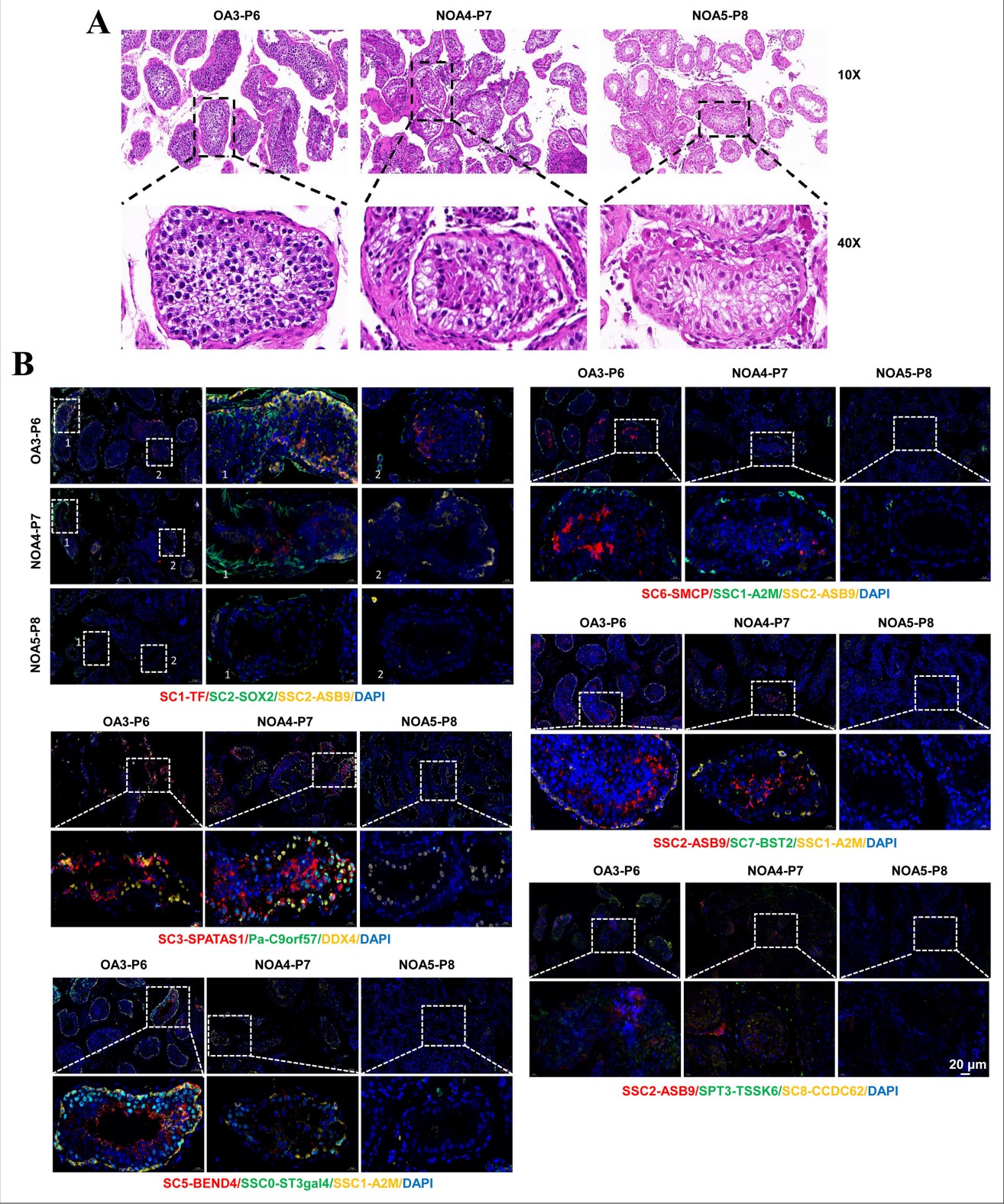

**Figure 7.** Co-localization of subpopulations of Sertoli cells and germ cells. (**A**) Representative images of HE staining of testicular tissues of obstructive azoospermia (OA) and non-obstructive azoospermia (NOA) patients. (**B**) Immunofluorescence analysis of germ cell and Sertoli cell subtype markers in testicular tissues. Blue DAPI indicates nuclei. The colors of the markers and cells indicate the fluorescent color of the corresponding antibody. The scale bar represents 20 μm.

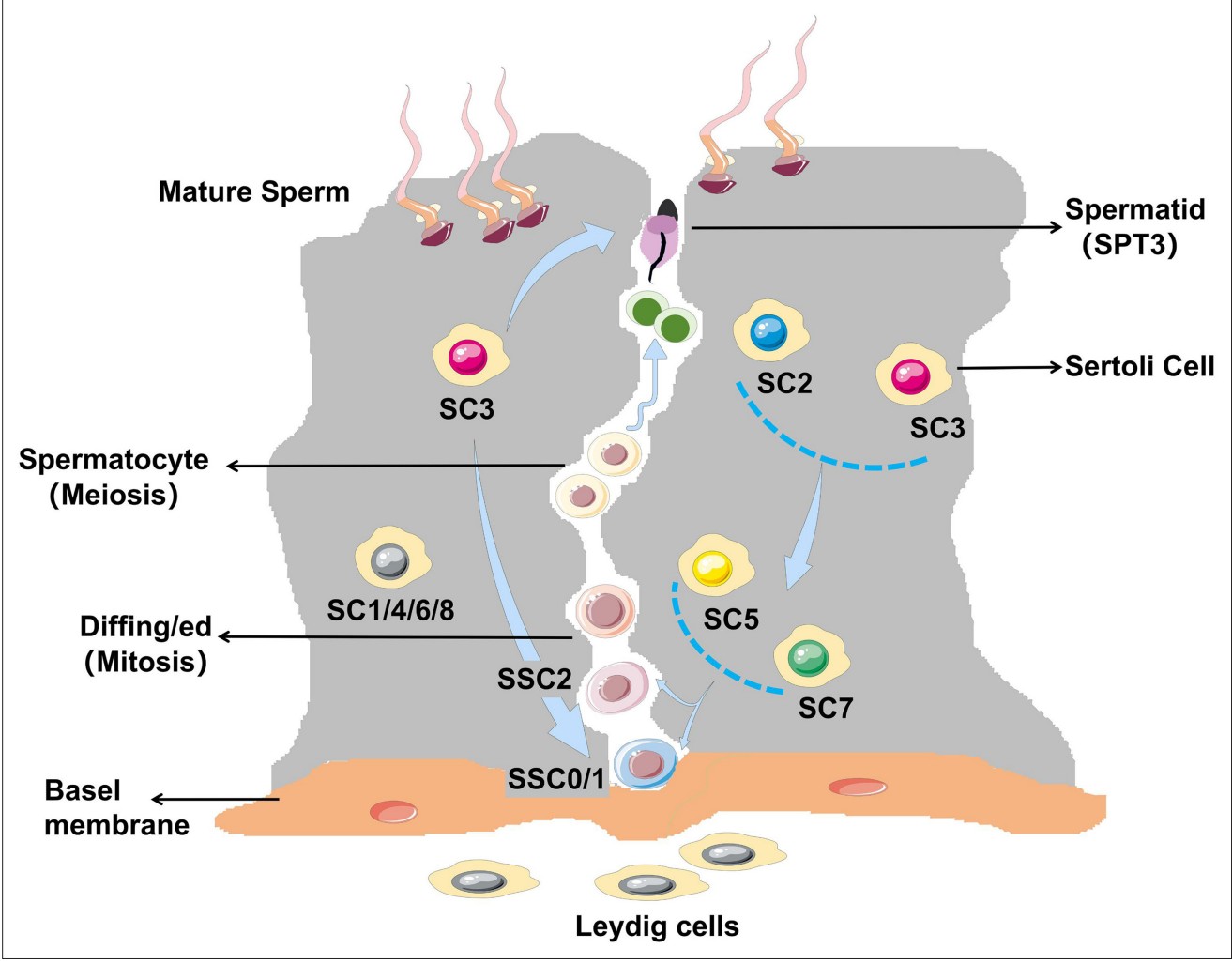

**Figure 8.** Schematic diagram of spermatogenesis and stage specificity of Sertoli cell subtypes.

## Discussion

NOA is due to pretesticular factors or testicular factors, usually abnormal spermatogenesis, which cannot be cured by surgery (*Zarezadeh et al., 2021*). Therefore, it is particularly important to study the etiology of NOA. Spermatogenesis is a complex developmental process that requires coordinated differentiation of multiple cell lines. In the present study, we performed scRNA-seq and scATAC-seq to assess the heterogeneity of germ cells and somatic cells. We identified 12 subtypes for germ cells, 8 subtypes for Sertoli cells, 12 subtypes for Leydig cells, 10 subtypes for PMCs, and 8 subtypes for macrophage and marker genes of specific cell type. The process of spermatogenesis was determined based on the cell trajectory analysis. Collectively, our results provide rich resources for exploring the potential mechanism of spermiogenesis.

In this study, some specific marker genes for germ cells were screened. We found that UTF1 and ID4 were specifically expressed in SSC1, and DMRT1 was specifically expressed in Diffing SPG. ID4 was a key regulator of SSC and ID4 also marked SSCs in the mouse testis (*Sun et al., 2015*). UTF1 was reported to be a marker for SSCs in stallions (*Jung et al., 2014*). UTF1 was discovered to play an important part in germ cell development, spermatogenesis, and male fertility in mice (*Raina et al., 2021*). DMRT1 is necessary for male sexual development (*Zhang et al., 2016*). The mechanism and expression pattern of these marker genes still need further research.

SSCs are adult stem cells in the testis of mammals that maintain spermatogenesis and are essential for male fertility, but the mechanisms remain elusive. UTF1 is a transcription factor expressed in SSC1, which plays a definite role in the proliferation and differentiation of pluripotent stem cells (*Tan*

and Wilkinson, 2019). Dmrt1 belongs to a family of conserved transcriptional regulators that control several key processes in mammalian testis, including germ cells and somatic cells. A recent discovery suggested that DMRT1 was essential for the formation of SSC and had gonadal-specific and amphoteric dimorphic expression patterns (Sohni et al., 2019). NANOS3 is a member of the highly conservative NANOS family. The decrease in NANOS3 expression could result in a decrease in the number of germ cells (Julaton and Reijo Pera, 2011). In this study, UTF1 was specifically expressed in SSC1, and DMRT1 and NANOS3 were the main specifically expressed genes in SSC2. Therefore, UTF1, DMRT1, and NANOS3 might promote SSC proliferation and were necessary for SSC differentiation.

Spermatogenesis consists of three processes: mitosis of spermatogonia, meiosis of spermatocyte, and deformation of spermatocyte. Abnormalities in any of these stages can result in spermatogenesis disorders. During this process, the testicular microenvironment in which spermatogenic cells are located plays an important role. In addition to Sertoli cells, the testicular microenvironment also includes myoid cells and mesenchymal cells and their secreted factors (Zhou et al., 2019). In this study, we clustered four somatic cell types: Sertoli cells, Leydig cells, PMCs, and macrophages. Sertoli cells play an important role in normal spermatogenesis (Zomer and Reddi, 2020). They provide the physical framework that supports the survival of germ cells and secretes unique growth factors and cytokines to assist in the development of germ cells (O'Donnell et al., 2022; Wu et al., 2020). In our study, the NOA3 group had the least number of sperm cells due to the lack of SC3/8, and the NOA2 group had a smaller number of SC3 cells than the normal OA group, so the number of sperm cells in the NOA2 group was between the NOA3 and OA groups. Although NOA2 did not have SC8 to participate in the formation of SPT3, this process was compensated by SC3, which was involved in cell formation throughout all stages of sperm development, and its function was particularly important. These data suggested that the association between germ cells and Sertoli cells was stage-specific, but this 'specificity' is not entirely isolated.

Notch receptors have been reported to play an important role in cell fate determination, maintenance, and differentiation of stem cells (Huang et al., 2013). Notch signaling components have been discovered to express in Sertoli and germ cells in the developing and mature testis (Garcia et al., 2013). In mice, a breakdown of Notch signaling can lead to abnormal cell differentiation and early embryonic lethality (McCright et al., 2001; Swiatek et al., 1994). Murta et al., 2014 also found notch signaling disrupts results in abnormal spermatogenesis in the mouse. Notch signaling was reported to be a key pathway regulating Sertoli cell physiology, and its changes may disturb reaction of Sertoli cells to androgens (Kamińska et al., 2020). Notch signaling in Sertoli cells is indirectly associated with germ cell development (Lu et al., 2019). The upregulation of notch signaling in Sertoli cells induced the transformation of the quiescence to differentiation and meiosis in germ cells (Garcia and Hofmann, 2013). In this study, we found that Notch signaling was involved in the interaction between germ cells and Sertoli cells. Therefore, Notch signaling might function in spermatogenesis.

This study has certain limitations. Firstly, the sample size is limited, which may not cover all subtypes of NOA, thereby affecting the universality of the conclusions. Additionally, this study primarily relies on scRNA-seq and scATAC-seq results, which are indirect evidence, and further experimental validation of the mechanisms is required. Finally, there are challenges in translating basic research into clinical applications, and individual differences may also affect the general applicability of treatment strategies. Therefore, future research needs to overcome these limitations to achieve a more comprehensive and in-depth understanding of the disease mechanisms and promote clinical applications.

In conclusion, our results revealed 12 germ cell subtypes and 8 Sertoli cell subtypes. We determined the process of spermatogenesis and found that SC3 subtypes (marked by SPATS1) of Sertoli cell played an important role in this process. The interaction between germ cells and Sertoli cells at each differentiation stage was stage-specific. The Notch1/2/3 signaling was discovered to be involved in germ cell–Sertoli cell interaction. Our results not only give us a comprehensive insight into human spermatogenesis, but also pave the way for determining molecules participated in the development of male germ cells, offering a powerful tool for further study on NOA.

# Materials and methods

## Human sample acquisition

Testicular tissues were obtained from three OA patients (OA1-P1, OA2-P2, OA3-P6) and five NOA patients (NOA1-P3, NOA2-P4, NOA3-P5, NOA4-P7, NOA5-P8) using micro-dissection of testicular sperm extraction separately. Testicular tissues obtained from OA1-P1, OA2-P2, NOA1-P3, NOA2-P4, and NOA3-P5 were used for scRNA-seq and scATAC-seq. Notably, the sperm concentration of the three NOA patients varied in descending order (NOA1>NOA2>NOA3), with NOA3 being the patient with complete absence of sperm. Patient clinical and laboratory characteristics are presented in *Supplementary file 5*. This study was approved by the Ethics Committee of School of Medicine, Renji Hospital, Shanghai Jiao Tong University (approval number: KY2020-193). All patients provided written informed consent.

## Sample dissociation

Testicular tissues were washed three times in Dulbecco's phosphate buffer saline (PBS) (Gibco, Waltham, MA, USA), and the tunica albugineas of testes were removed. Testicular tissues were cut into small pieces of 1–2 mm and digested in 10 mL Dulbecco's Modified Eagle's Medium: F12 media (Gibco) containing 1 mg/mL type IV collagenase for 15 min on a rotor at 37°C. Subsequently, tubules were washed again and incubated with 10 mL F12 media containing 500 µg/mL DNase I and 200 µg/mL trypsin for 15 min at 37°C. The digestion was stopped with the addition of fetal bovine serum (FBS) (Gibco). The single cells were obtained by filtering through a 40 µm cell strainer. The cells were resuspended in 1 mL Dulbecco's PBS and 5 mL red blood cell lysis buffer to remove red blood cells. Then, the cells were washed with DPBS and cryopreserved in DPBS with 1% FBS (Gibco) until further use. For scRNA-seq and scATAC-seq assay, the cryopreserved cells were thawed and incubated with a Dead Cell Removal Kit (Miltenyi Biotec) to clear dead/stressed cells in accordance with the manufacturer's instruction.

## scRNA-seq library preparation

scRNA-seq was carried out using 10x Chromium Single Cell 3' Reagent according to the manufacturer's instruction. In brief, viable single cells were resuspended in PBS with 0.04% bovine serum albumin (BSA; Sigma) and counted using the hemocytometer. Suspensions containing 5000–8000 cells per sample were mixed with the RT-PCR reaction and loaded into the Single Cell Chip B and processed through the 10× controller for droplet production. Then, in-drop lysis and reverse transcription occur and mRNA transcripts from single cells were barcoded to determine the cell origin. Following reverse transcription, barcoded cDNAs were purified, amplified by 12 cycles of PCR, end-repaired, and ligated with Illumina adapters. The final libraries were sequenced on the Illumina NovaSeq 6000 platform with paired end 150 bp sequencing. Each sample was sequenced with depth of 100 G of raw reads.

## scRNA-seq data processing and analysis

The scRNA-seq data produced by Illumina NovaSeq 6000 sequencing were processed and mapped to human reference genome hg38. After mapping, the outputs were processed using the Seurat package in R. To filter out low-quality cells, we referred to the following criteria: cells with few genes per cell (<50) or plenty of molecules per cell (>20,000); cells with over 30% of mitochondrial genes. Normalization was carried out in accordance with the package manual (https://satijalab.org/seurat/v3.1/). We identified cell clusters with Uniform Manifold Approximation and Projection (UMAP). The differentially expressed genes were analyzed using the FindMarkers function. Gene set enrichment analysis was carried out using the clusterProfiler R package. Regulatory gene network analysis was carried out using SCENIC.

## Pseudotime analysis

We applied the Single-Cell Trajectories analysis utilizing Monocle2 (http://cole-trapnell-lab.github.io/monocle-release) using DDR-Tree and default parameter. Before Monocle analysis, we selected marker genes of the Seurat clustering result and raw expression counts of the cell passed filtering. Based on the pseudotime analysis, branch expression analysis modeling (BEAM Analysis) was applied for branch fate-determined gene analysis.

## Velocity analysis

RNA velocity analysis was conducted using scVelo's (version 0.2.1) generalized dynamical model. The spliced and unspliced mRNA were quantified using Velocity (version 0.17.17).

## Cell cycle analysis

To quantify the cell cycle phases for individual cells, we employed the CellCycleScoring function from the Seurat package. This function computes cell cycle scores using established marker genes for cell cycle phases. Cells showing a strong expression of G2/M-phase or S-phase markers were designated as G2/M-phase or S-phase cells, respectively. Cells that did not exhibit significant expression of markers from either category were classified as G1-phase cells.

## Cell communication analysis

To enable a systematic analysis of cell–cell communication molecules, we applied cell communication analysis based on the CellPhoneDB, a public repository of ligands, receptors, and their interactions. Membrane, secreted, and peripheral proteins of the cluster of different time point were annotated. Significant mean and cell communication significance (p-value<0.05) was calculated based on the interaction and the normalized cell matrix achieved by Seurat Normalization.

## SCENIC analysis

To assess transcription factor regulation strength, we applied the single-cell regulatory network inference and clustering (pySCENIC, version 0.9.5) workflow using the 20,000 motifs database for RcisTarget and GRNboost.

## scATAC-seq library preparation

scATAC-seq was conducted using 10x Chromium Single Cell ATAC Reagent (V1.1 chemistry) in accordance with the manufacturer's instruction. In short, cells were resuspended in PBS with 0.04% BSA (Sigma) and counted. Next, cells were cultured with cold lysis buffer on ice for 5 min. Tn5 transposase reaction was performed using the Tagment DNA Enzyme 1 (Illumina) at 37°C for 30 min. Cells were processed for targeting a recovery of 10,000 cells per sample, and amplification was performed with 12 cycles of PCR for library construction. The libraries were sequenced on Illumina NovaSeq 6000 platform, and each sample was sequenced with a depth of 200M raw reads.

## scATAC-seq data preprocessing and analysis

The scATAC-seq data analysis was performed as previously described. In brief, the sequence reads were mapped to the human reference genome hg38. The R package Signac was used to process the generated outputs. The Latent Semantic indexing algorithm, T-distributed Stochastic Neighbor Embedding algorithm, and Uniform Manifold Approximation algorithm were used for dimension reduction and information display. Each motif of the cell can be annotated by associating identified open chromatin sites with the JASPAR database. By using CHIPseeker, the distribution of peak in different functional regions of the genome was annotated and statistically analyzed. Prediction of results for each cell type in scATAC-seq can be achieved by combining scRNA-seq data with scATAC-seq using KNN.

## Integrating scATAC-seq and scRNA-seq analysis

We primarily utilized Seurat for scRNA-seq data processing and Signac (version 1.13.0) for scATAC-seq data analysis. To establish connections between scRNA-seq and scATAC-seq datasets, we approximated gene transcriptional activity by measuring ATAC-seq signals within 2 kb upstream and the gene body regions using Signac's GeneActivity function. These gene activity scores from scATAC-seq data, combined with gene expression levels from scRNA-seq, were input for canonical correlation analysis. Subsequently, we pinpointed anchors between the two datasets with the FindTransferAnchors function. We proceed to annotate scATAC-seq cells. For joint visualization in a UMAP plot, we projected RNA expression onto scATAC-seq cells using the pre-established anchors and integrated the datasets.

## Immunofluorescence

The testicular tissues were fixed in 4% paraformaldehyde at room temperature for 30 min, permeabilized with 0.5% Triton X-100, and blocked with 1% BSA (Gibco). The samples were incubated

overnight at 4°C with primary antibodies, including ST3GAL4 (13546-1-AP, Proteintech), A2M (ab109422, Abcam), DDX4 (ab270534, Abcam), ASB9 (22728, SAB), TEX19 (AF6319, R&D), TSSK6 (H00083983-M02, Novus), PRAP1 (11932-1-AP, Proteintech), BST2 (H00000684-B02P, Novus), SOX9 (ab185966, Abcam), CCDC62 (25981-1-AP, Proteintech), TF (ab185966, Abcam), SOX2 (ab92494, Abcam), SPATS1 (NBP2-31037, Novus), C9orf57 (orb156207, Biorbyt), BEND4 (24711-1-AP, Proteintech), and SMCP (NBP1-81252, Novus). After washing with PBS, the samples were incubated with a secondary antibody (Abcam) for 1 h at room temperature. DAPI (Cell Signaling, MA, USA) was used to stain cell nucleus. After washing three times with PBS, the sections were performed for photograph under a fluorescence microscope (Carl Zeiss, Oberkochen, Germany).

## Acknowledgements

The research was supported by the ShenZhen Science and Technology Program (grant no. JCYJ20230807111304010) and Futian Healthcare Research Project (grant no. FTWS2023021).

## Additional information

### Funding

| Funder | Grant reference number | Author |
| --- | --- | --- |
| ShenZhen Science and Technology Program | JCYJ20230807111304010 | Shimin Wang |
| Futian Healthcare Research Project | FTWS2023021 | Dong Zhao |

The funders had no role in study design, data collection and interpretation, or the decision to submit the work for publication.

### Author contributions

Shimin Wang, Hongxiang Wang, Conceptualization, Data curation, Writing – original draft; Bicheng Jin, Qingliang Zheng, Formal analysis, Visualization; Hongli Yan, Methodology, Project administration; Dong Zhao, Supervision, Project administration, Writing – review and editing

### Author ORCIDs

Shimin Wang ⬤ https://orcid.org/0009-0000-8744-2810
Qingliang Zheng ⬤ http://orcid.org/0000-0002-5626-8423
Dong Zhao ⬤ http://orcid.org/0000-0002-2118-2910

### Ethics

This study was approved by the Ethics Committee of Renji Hospital, Shanghai Jiao Tong University School of Medicine [approval number: KY2020-193]. All patients provided written informed consent.

Reviewer #1 (Public review): https://doi.org/10.7554/eLife.97958.3.sa1
Reviewer #2 (Public review): https://doi.org/10.7554/eLife.97958.3.sa2
Reviewer #3 (Public review): https://doi.org/10.7554/eLife.97958.3.sa3
Author response https://doi.org/10.7554/eLife.97958.3.sa4

## Additional files

### Supplementary files

Supplementary file 1. The number of different kinds of cells in five samples in scRNA-seq.
Supplementary file 2. The number of different kinds of cells in five samples in scATAC-seq.
Supplementary file 3. The number of Sertoli cell subtypes in five samples in scRNA-seq.
Supplementary file 4. The number of Sertoli cell subtypes in five samples in scATAC-seq.
Supplementary file 5. Patient characteristics in this study.

MDAR checklist

## Data availability

ScRNA-seq data have been deposited in the NCBI Gene Expression Omnibus with the accession number GSE202647, and scATAC-seq data have been deposited in the NCBI database with the accession number PRJNA1177103. Other sequencing datasets used can be found in NCBI GEO under accession numbers GSE149512.

The following datasets were generated:

| Author(s) | Year | Dataset title | Dataset URL | Database and Identifier |
| --- | --- | --- | --- | --- |
| Wang S, Wang H, Jin B, Zhao D | 2022 | Single-cell analysis reveals human spermatogenesis and germ cell-sertoli cell interaction | https://www.ncbi.nlm.nih.gov/geo/query/acc.cgi?acc=GSE202647 | NCBI Gene Expression Omnibus, GSE202647 |
| Wang S | 2024 | ScRNA-seq and scATAC-seq reveal that sertoli cells mediate spermatogenesis disorders through stage-specific communications in non-obstructive azoospermia | https://www.ncbi.nlm.nih.gov/bioproject/PRJNA1177103/ | NCBI BioProject, PRJNA1177103 |

The following previously published dataset was used:

| Author(s) | Year | Dataset title | Dataset URL | Database and Identifier |
| --- | --- | --- | --- | --- |
| Zhao L, Yao C, Zhou Z, Li Z | 2020 | Single-cell analysis of developing and azoospermia human testicles reveals central role of Sertoli cells | https://www.ncbi.nlm.nih.gov/geo/query/acc.cgi?acc=GSE149512 | NCBI Gene Expression Omnibus, GSE149512 |

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
