## [Editor Report · eLife Assessment]

This study provides **valuable** scRNA-seq and scATAC-seq data for testicular tissues from patients with spermatogenesis disorders. By examining the transcriptomic and epigenetic changes in Sertoli cells, the authors uncovered key regulatory mechanisms underlying male infertility and identified potential therapeutic targets. While some of the cellular profiling results are **convincing**, the analyses for differential profiling of NOA cases and epigenomics data remain **incomplete**.

---

## [Referee Report · Reviewer #1 (Public review)]

Summary:

In this study, Wang and colleagues generate single-cell transcriptome and chromatin accessibility data from testicular tissues of two OA and three NOA cases. The authors analyze this dataset to identify novel cellular populations, marker genes, and inter-population interactions that may contribute to proper spermatogenesis. Then they propose a role of specific Sertoli cell subtypes and their interactions via Notch signaling in germ cell development. However, I remain skeptical of their central argument (also highlighted in the title) that stage-specific interactions between Sertoli and germ cells are a key component in NOA development, as my initial concerns regarding potential data misrepresentation, lack of statistical testing, and the rationale behind some of the analyses have not been sufficiently addressed.

(1) As noted in my previous comments, the analysis of Sertoli cell subtypes is potentially misleading and lacks proper statistical support. The authors claim a significant loss of Sertoli subpopulations in NOA cases, and provide the absolute number of cells in Figure 6B. However, this observation could easily be driven by the total number of cells captured during the experiment and the anatomical location of the specimens. There is no statistical basis to make the claim that this loss is significant. Furthermore, the same analysis should be performed on scATAC-seq cells and presented alongside.

(2) As pointed out in my initial concerns, some parts of the analyses require additional explanation to clarify their logical flow. For example, the logic of using between-sample correlations to assess colocalization of Sertoli and germ cells is lost on me. How can this be used to infer the important role of specific Sertoli cell populations in spermatogenesis, other than the fact that some of the genes are more co-expressed in the sub-populations? And how is this related to the claim that these cell populations are actually co-localized in the tissue? The authors then dedicate nearly a page describing the pathways enriched in Sertoli and germ cells, but the relevance is unclear, and the argument that these subtypes are functionally related is not convincing enough.

(3) The statement regarding Notch signaling as a critical component in Sertoli and germ cell interaction is not supported by actual evidence. The inference based on CellphoneDB and an epigenome snapshot that shows not much difference are insufficient to justify this claim.

(4) The manuscript is overly wordy and descriptive, making it difficult to read and understand the points. The main text needs to be more concise and on point, with unnecessary details removed to sharpen the key points. Non-essential results (e.g. Figure S10 and S11) unrelated to the main argument should be removed.

---

## [Referee Report · Reviewer #2 (Public review)]

Summary:

Shimin Wang et al. investigated the role of Sertoli cells in mediating spermatogenesis disorders in non-obstructive azoospermia (NOA) through stage-specific communications. The authors utilized scRNA-seq and scATAC-seq to analyze the molecular and epigenetic profiles of germ cells and Sertoli cells at different stages of spermatogenesis.

Strengths:

By understanding the gene expression patterns and chromatin accessibility changes in Sertoli cells, the authors sought to uncover key regulatory mechanisms underlying male infertility and identify potential targets for therapeutic interventions. They emphasized that the absence of the SC3 subtype would be a major factor contributing to NOA.

Comments on revisions:

The authors have addressed my concerns. I have no further comments.

---

## [Referee Report · Reviewer #3 (Public review)]

Summary:

This study profiled the single-cell transcriptome of human spermatogenesis and provided many potentials molecular markers for developing testicular puncture specific marker kits for NOA patients.

Strengths:

Perform single-cell RNA sequencing (scRNA-seq) and single-cell assay for transposase-accessible chromatin sequencing (scATAC-seq) on testicular tissues from two OA patients and three NOA patients

Weaknesses:

Most results are analytical and lack specific experiments to support these analytical results and hypotheses.

Comments on revisions:

In the revised version of the manuscript, the authors made some effort to revise their manuscript according to reviewers' comments and addressed the problems that I had raised before.

I have no other serious criticisms regarding the revised manuscript.

---

## [Author Response]

The following is the authors’ response to the original reviews.

**Public Reviews:**

**Reviewer #1 (Public Review):**
Summary:The manuscript is dedicated heavily to cell type mapping and identification of sub-type markers in the human testis but does not present enough results from cross-investigation between NOA cases versus control. Their findings are mostly based on transcriptome and the authors do not make enough use of the scATAC-seq data in their analyses as they put forward in the title. Overall, the authors should do more to include the differential profile of NOA cases at the molecular level - specific gene expression, chromatin accessibility, TF binding, pathway, and signaling that are perturbed in NOA patients that may be associated with azoospermia.Strengths:(1) The establishment of single-cell data (both RNA and ATAC) from the human testicular tissues is noteworthy.(2) The manuscript includes extensive mapping of sub-cell populations with some claimed as novel, and reports marker gene expression.(3) The authors present inter-cellular cross-talks in human testicular tissues that may be important in adequate sperm cell differentiation.Weaknesses:(1) A low sample size (2 OA and 3 NOA cases). There are no control samples from healthy individuals.

Thank you for your comments. We recognize that the small sample size in this study somewhat limits its generalizability. However, in transcriptomic research, limited sample sizes are a common issue due to the complexities involved in acquiring samples, particularly in studies about the reproductive system. Healthy testicular tissue samples are difficult to obtain, and studies (doi: 10.18632/aging.203675) have used obstructive azoospermia as a control group in which spermatogenesis and development are normal.

(2) Their argument about interactions between germ and Sertoli cells is not based on statistical testing.

Thank you for your comments. Due to limited funding, we have not yet fully and deeply conducted validation experiments, but we plan to carry out related experiments in the later stage. We hope that the publication of this study will help to obtain more financial support to further investigate the interactions between germ cells and Sertoli cells.

(3) Rationale/logic of the study. This study, in its present form, seems to be more about the role of sub-Sertoli population interactions in sperm cell development and does not provide enough insights about NOA.

Thank you for your comments. In Figure 6, we conducted an in-depth analysis and comparison of the differences between the Sertoli cell subtypes and the germ cell subtypes involved in spermatogenesis in the OA and NOA groups. The results revealed that in the NOA group, especially in the NOA3 group, which has a lower sperm count compared to NOA2 and NOA1, there is a significant loss of Sertoli cell subtypes including SC3, SC4, SC5, SC6, and SC8. The NOA1 group, with a sperm count close to that of the OA group, also had a Sertoli cell profile similar to the OA group. The NOA2 group, with a sperm count between that of NOA1 and NOA3, also exhibited an intermediate profile of Sertoli cell subtypes. Therefore, we suggest that change in Sertoli cell subtypes is a key factor affecting sperm count, rather than just the total number of Sertoli cells. We believe that through these analyses, we can provide in-depth insights into NOA, and we hope that the publication of this study will help obtain more funding support to further validate and expand on these findings.

(4) The authors do not make full use of the scATAC-seq data.

Thank you for your comments.We have added analysis of the scATAC-seq data and shown in the revised manuscript.

**Reviewer #2 (Public Review):**
Summary:Shimin Wang et al. investigated the role of Sertoli cells in mediating spermatogenesis disorders in non-obstructive azoospermia (NOA) through stage-specific communications. The authors utilized scRNA-seq and scATAC-seq to analyze the molecular and epigenetic profiles of germ cells and Sertoli cells at different stages of spermatogenesis.Strengths:By understanding the gene expression patterns and chromatin accessibility changes in Sertoli cells, the authors sought to uncover key regulatory mechanisms underlying male infertility and identify potential targets for therapeutic interventions. They emphasized that the absence of the SC3 subtype would be a major factor contributing to NOA.Weaknesses:Although the authors used cutting-edge techniques to support their arguments, it is difficult to find conceptual and scientific advances compared to Zeng S et al.'s paper (Zeng S, Chen L, Liu X, Tang H, Wu H, and Liu C (2023) Single-cell multi-omics analysis reveals dysfunctional Wnt signaling of spermatogonia in non-obstructive azoospermia. Front. Endocrinol. 14:1138386.). Overall, the authors need to improve their manuscript to demonstrate the novelty of their findings in a more logical way.

Thank you for your detailed review of our work. We greatly appreciate your feedback and have made revisions to our manuscript accordingly.

Regarding the novelty of our research, we believe our study offers conceptual and scientific advances in several ways:

We have systematically revealed the stage-specific roles of Sertoli cell subtypes in different stages of spermatogenesis, particularly emphasizing the crucial role of the SC3 subtype in non-obstructive azoospermia (NOA). Additionally, we identified that other Sertoli cell subtypes (SC1, SC2, SC3...SC8, etc.) also collaborate in a stage-specific manner with different subpopulations of spermatogenic cells (SSC0, SSC1/SSC2/Diffed, Pa...SPT3). These findings provide new insights into the understanding of spermatogenesis disorders.

Compared to the study by Zeng S et al., our research not only focuses on the functional alterations in Sertoli cells but also comprehensively analyzes the interaction patterns between Sertoli cells and spermatogenic cells using scRNA-seq and scATAC-seq technologies. We uncovered several novel regulatory networks that could serve as potential targets for the diagnosis and treatment of NOA.

We sincerely appreciate your constructive comments and will continue to explore this area further, aiming to make a more significant contribution to the understanding of NOA mechanisms.

**Reviewer #3 (Public Review):**
Summary:This study profiled the single-cell transcriptome of human spermatogenesis and provided many potential molecular markers for developing testicular puncture-specific marker kits for NOA patients.Strengths:Perform single-cell RNA sequencing (scRNA-seq) and single-cell assay for transposase-accessible chromatin sequencing (scATAC-seq) on testicular tissues from two OA patients and three NOA patients.Weaknesses:Most results are analytical and lack specific experiments to support these analytical results and hypotheses.

Thank you for your thorough review of our work. We highly value your feedback and have made revisions to our manuscript accordingly. Indeed, we have conducted immunofluorescence (IF) experiments to validate the data obtained from single-cell sequencing and have expanded the sample size to enhance the reliability of our results. To better present these validation experiments, we have reorganized and renamed the sample information, making it easier for you to understand which samples were used in the specific experiments. Following the publication of this paper, we plan to secure additional funding to deepen our research, particularly in the area of experimental validation. We sincerely appreciate your support and insightful suggestions, which have greatly helped guide our future research directions.

**Reviewer #1 (Recommendations For The Authors):**
(1) The authors should include results from cross-investigation comparing NOA/OA patients versus controls.

Thank you for your comments. In this study, OA was the control group. Healthy testicular tissue samples are difficult to obtain, and studies (doi: 10.18632/aging.203675) have used OA as a control group in which spermatogenesis and development are normal.

(2) In Table S1, the authors should also include the metric for scATAC-seq, and do more to show the findings the authors obtained in RNA is replicated with chromatin accessibility.

Thank you for your comments. We have added Table S2, which includes the metric for scATAC-seq.

(3) A single sample from each OA and NOA group may not be enough to confirm colocalization. The authors should include results from all available samples and use quantitative measures.

Thank you for your comments. I apologize that the sample size in this study was less than three and we could not conduct quantitative analysis. We will increase the sample size and conduct corresponding experiments in subsequent research.

(4) The Methods section does not include enough description to follow how the analyses were carried out, and is missing information on some of the key procedures such as velocity and cell cycle analyses.

Thank you for your comments. The method about velocity and cell cycle analyses was added in the revised manuscript. The description is as follows:

“Velocity analysis

RNA velocity analysis was conducted using scVelo's (version 0.2.1) generalized dynamical model. The spliced and unspliced mRNA was quantified by Velocity (version 0.17.17).”

“Cell cycle analysis

To quantify the cell cycle phases for individual cell, we employed the CellCycleScoring function from the Seurat package. This function computes cell cycle scores using established marker genes for cell cycle phases as described in a previous study by Nestorowa et al. (2016). Cells showing a strong expression of G2/M-phase or S-phase markers were designated as G2/M-phase or S-phase cells, respectively. Cells that did not exhibit significant expression of markers from either category were classified as G1-phase cells.”

(5) For the purpose of transparency, the authors should upload codes used for analyses so that each figure can be reproduced. All raw and processed data should be made publicly available.

Thank you for your comments. We have deposited scRNA-seq and scATAC-seq data in NCBI. ScRNA-seq data have been deposited in the NCBI Gene Expression Omnibus with the accession number GSE202647, and scATAC-seq data have been deposited in the NCBI database with the accession number PRJNA1177103.

**Reviewer #2 (Recommendations For The Authors):**
The detailed points the authors need to improve are attached below.The results presented in the study have several weaknesses:In Figure 1A, it's required to show HE staining results of all patients who underwent single-cell analysis were provided.

Thank you very much for your valuable suggestions. In Figure 1, we present the HE staining results paired with the single-cell data, covering all patients involved in the single-cell analysis.

- Saying "identification of novel potential molecular markers for distinct cell types" seems unsupported by the data.

Thank you for your comments. I'm sorry for the inaccuracy of my description. We have revised this sentence. The description is as follows: These findings indicate that the scRNA-seq data from this study can serve for cellular classification.

- The methods suggest an integrated analysis of scRNA-seq and scATAC-seq, but from the figures, it seems like separate analyses were performed. It's necessary to have data showing the integrated analysis.

Thank you for your comments. We have added an integrated analysis of scRNA-seq and scATAC-seq. The results were shown in Figure S2.

Figure 2 does not seem to well cover the diversity of germ cell subtypes. The main content appears to be about the differentiation process, and it seems more focused on SSCs (stem cell types), but the intended message is not clearly conveyed.

Thank you for your comments. Figure S1 revealed the diversity of germ cell subtypes. The second part of the results described the integrated findings from Figures 2 and S1.

- In Figure 2B, pseudotime could be shown, and I wonder if the pseudotime in this analysis shows a similar pattern as in Figure 2D.

Thank you for your comments. Figure 2B revealed the pseudotime analysis of 12 germ cell subpopulation. Figure 2D revealed RNA velocity of 12 germ cell subpopulation. The two methods are both used for cell trajectory analysis. The pseudotime in Figure 2B showed a similar pattern as in Figure 2D.

- While staining occurs within one tissue, saying they are co-expressed seems inaccurate as the staining locations are clearly distinct. For example, the staining patterns of A2M and DDX4 (a classical marker) are quite different, so it's hard to claim A2M as a new potential marker just because it's expressed. Also, TSSK6 was separately described as having a similar expression pattern to DDX4, but from the IF results, it doesn't seem similar.

Thank you for your comments. We have revised the Figure.

- It was described that A2M (expressed in SSC0-1), and ASB9 (expressed in SSC2) have open promoter sites in SSC0, SSC2, and Diffing_SPG, but it doesn't seem like they are only open in the promoters of those cell types. For example, there doesn't seem to be a peak in Diffing for either gene. The promoter region of the tracks is not very clear, so overall figure modification seems necessary.

Thank you for your comments. We have revised the Figure.

- The ATAC signal scale for each genomic region should be included, and clear markings for the TSS location and direction of the genes are needed.

Thank you for your comments. We have revised the figure and shown in the revised manuscript.

Figure 3A mostly shows the SSC2 in the G2/M phase, so it seems questionable to call SSC0/1 quiescent. Also, I wonder if the expression of EOMES and GFRA1 is well distinguished in the SSC subtypes as expected.

Thank you for your comments. We will validate in subsequent experiments whether the expression of EOMES and GFRA1 is clearly distinguished in the SSC subtypes.

- In Figure 3C, it would be good to have labels indicating what the x and y axes represent. The figure seems complex, and the description does not seem to fully support it.

Thank you for your comments. We have added labels indicating what the x and y axes represent in the Figure 3C. The x and y axes represent spliced and unspliced mRNA ratios, respectively.

- While TFs are the central focus, it's disappointing that scATAC-seq was not used.

Thank you for your comments. TFs analysis using scATAC-seq will be carried out in the future.

Figure 4: It would be good to have a more detailed discussion of the differences between subtypes, such as through GO analysis. The track images need modification like marking the peaks of interest and focusing more on the promoter region, similar to the previous figures.

Thank you for your comments. GO analysis results were put in Figure S5. The description is as follows:

As shown in Figure S5, SC1 were mainly involved in cell differentiation, cell adhesion and cell communication; SC2 were involved in cell migration, and cell adhesion; SC3 were involved in spermatogenesis, and meiotic cell cycle; SC4 were involved in meiotic cell cycle, and positive regulation of stem cell proliferation; SC5 were involved in cell cycle, and cell division; SC6 were involved in obsolete oxidation−reduction process, and glutathione derivative biosynthetic process; SC7 were involved in viral transcription and translational initiation; SC8 were involved in spermatogenesis and sperm capacitation.

In Figure 5, it would be good to have criteria for the novel Sertoli cell subtype presented. CCDC62 is presented as a representative marker for the SC8 cluster, but from Figure 4C, it seems to be quite expressed in the SC3 cluster as well. Therefore, in Figure 5E's protein-level check, it's unclear if this truly represents a novel SC8 subtype.

Thank you for your comments. CCDC62 expression was higher in SC8 cluster than in SC3. Since some molecular markers were not commercially available in the market, CCDC62 was selected as SC8 marker for immunofluorescence verification. Immunofluorescence results showed that CCDC62 is a novel SC8 marker.

- It might have been more meaningful to use SOX9 as a control and show that markers in the same subtype are expressed in the same location.

Thank you for your comments. To determine PRAP1, BST2, and CCDC62 as new markers for the SC subtype, we co-stained them with SOX9 (a well-known SC marker).

- Figures 4 and 5 could potentially be combined into one figure.

Thank you for your comments. Since combining Figures 4 and 5 into a single image would cause the image to be unclear, two images are used to show it.

In Figure 6, it would be good to support the results with more NOA patient data.

Thank you for your comments. Patient clinical and laboratory characteristics has been presented in Table 1.

- Rather than claiming the importance of SC3 based on 3 single-cell patient data, it would be better to validate using public data with SC3 signature genes (e.g., showing the correlation between germ cell and SC3 ratios).

Thank you for your comments. I'm sorry I didn't find public data with SC3 signature genes. In the future, we will verify the importance of SC3 through in vivo and in vitro experiments.

- 462: It seems to be referring to Figure 6G, not 6D.

Thank you for your comments. We have revised it. The description is as follows: As shown in Figure 6G, State 1 SC3/4/5 were tended to associated with PreLep, SSC0/1/2, and Diffing and Diffed-SPG sperm cells (R > 0.72).

In Figure 7, the spermatogenesis process is basically well-known, so it would be better to emphasize what novel content is being conveyed here. Additionally, emphasizing the importance of SC3 in the overall process based on GO results leaves room for a better approach.

Thank you for your valuable suggestions. Regarding Figure 7, we recognize that the spermatogenesis process is well-known, and we will focus on highlighting the novel content, particularly the role and significance of the SC3 subtype in spermatogenesis disorders. As for the importance of SC3 in the overall process based on GO results, we have validated this in Figure 8 through co-staining experiments between Sertoli cells and spermatogenic cells in OA and NOA groups. The results demonstrate a significant correlation between the number of SC3-positive cells and SPT3 spermatogenic cells, particularly in the NOA5-P8 group, where both SC3 and SPT3 cell counts are notably lower than in the NOA4-P7 group. This further supports the critical role of SC3 in the spermatogenesis process. Your suggestions have prompted us to refine our data presentation and more clearly emphasize the novel aspects of our research. We will continue to strive to ensure that every part of our research contributes meaningfully to the academic community. Thank you again for your guidance.

In Figure 8, only the contents of the IF-stained proteins are listed, which seems slightly insufficient to constitute a subsection on its own. It might have been better to conclude by emphasizing some subtypes.

Thank you for your comments. We have combined this part of the results with other results into one section. The description is as follows:

“Co-localization of subpopulations of Sertoli cells and germ cells

To determine the interaction between Sertoli cells and spermatogenesis, we applied Cell-PhoneDB to infer cellular interactions according to ligand-receptor signalling database. As shown in Figure 6G, compared with other cell types, germ cells were mainly interacted with Sertoli cells. We futher performed Spearman correlation analysis to determine the relationship between Sertoli cells and germ cells. As shown in Figure 6H, State 1 SC3/4/5 were tended to be associated with PreLep, SSC0/1/2, and Diffing and Diffed-SPG sperm cells (R > 0.72). Interestingly, SC3 was significantly positively correlated with all sperm subpopulations (R > 0.5), suggesting an important role for SC3 in spermatogenesis and that SC3 is involved in the entire process of spermatogenesis. Subsequently, to understand whether the functions of germ cells and Sertoli cells correspond to each other, GO term enrichment analysis of germ cells and sertoli cells was carried out (Figure S3, S4). We found that the functions could be divided into 8 categories, namely, material energy metabolism, cell cycle activity, the final stage of sperm cell formation, chemical reaction, signal communication, cell adhesion and migration, stem cells and sex differentiation activity, and stress reaction. These different events were labeled with different colors in order to quickly capture the important events occurring in the cells at each stage. As shown in Figure S3, we discovered that SSC0/1/2 was involved in SRP-dependent cotranslational protein targeting to membrane, and cytoplasmic translation; Diffing SPG was involved in cell division and cell cycle; Diffied SPG was involved in cell cycle and RNA splicing; Pre-Leptotene was involved in cell cycle and meiotic cell cycle; Leptotene_Zygotene was involved in cell cycle and meiotic cell cycle; Pachytene was involved in cilium assembly and spermatogenesis; Diplotene was involved in spermatogenesis and cilium assembly; SPT1 was involved in cilium assembly and flagellated sperm motility; SPT2 was involved in spermatid development and flagellated sperm motility; SPT3 was involved in spermatid development and spermatogenesis. As shown in Figure S4, SC1 were mainly involved in cell differentiation, cell adhesion and cell communication; SC2 were involved in cell migration, and cell adhesion; SC3 were involved in spermatogenesis, and meiotic cell cycle; SC4 were involved in meiotic cell cycle, and positive regulation of stem cell proliferation; SC5 were involved in cell cycle, and cell division; SC6 were involved in obsolete oxidation−reduction process, and glutathione derivative biosynthetic process; SC7 were involved in viral transcription and translational initiation; SC8 were involved in spermatogenesis and sperm capacitation. The above analysis indicated that the functions of 8 Sertoli cell subtypes and 12 germ cell subtypes were closely related.

To further verify that Sertoli cell subtypes have "stage specificity" for each stage of sperm development, we firstly performed HE staining using testicular tissues from OA3-P6, NOA4-P7 and NOA5-P8 samples. The results showed that the OA3-P6 group showed some sperm, with reduced spermatogenesis, thickened basement membranes, and a high number of sertoli cells without spermatogenic cells. The NOA4-P7 group had no sperm initially, but a few malformed sperm were observed after sampling, leading to the removal of affected seminiferous tubules. The NOA5-P8 group showed no sperm in situ (Figure 7A). Immunofluorescence staining in Figure 7B was performed using these tissues for validation. ASB9 (SSC2) was primarily expressed in a wreath-like pattern around the basement membrane of testicular tissue, particularly in the OA group, while ASB9 was barely detectable in the NOA group. SOX2 (SC2) was scattered around SSC2 (ASB9), with nuclear staining, while TF (SC1) expression was not prominent. In NOA patients, SPATS1 (SC3) expression was significantly reduced. C9orf57 (Pa) showed nuclear expression in testicular tissues, primarily extending along the basement membrane toward the spermatogenic center, and was positioned closer to the center than DDX4, suggesting its involvement in germ cell development or differentiation. BEND4, identified as a marker fo SC5, showed a developmental trajectory from the basement membrane toward the spermatogenic center. ST3GAL4 was expressed in the nucleus, forming a circular pattern around the basement membrane, similar to A2M (SSC1), though A2M was more concentrated around the outer edge of the basement membrane, creating a more distinct wreath-like arrangement. In cases of impaired spermatogenesis, this arrangement becomes disorganized and loses its original structure. SMCP (SC6) was concentrated in the midpiece region of the bright blue sperm cell tail. In the OA group, SSC1 (A2M) was sparsely arranged in a rosette pattern around the basement membrane, but in the NOA group, it appeared more scattered. SSC2 (ASB9) expression was not prominent. BST2 (SC7) was a transmembrane protein primarily localized on the cell membrane. In the OA group, A2M (SSC1) was distinctly arranged in a wreath-like pattern around the basement membrane, with expression levels significantly higher than ASB9 (SSC2). TSSK6 (SPT3) was primarily expressed in OA3-P6, while CCDC62 (SC8) was more abundantly expressed in NOA4-P7, with ASB9 (SCC2) showing minimal expression. Taken together, germ cells of a particular stage tended to co-localize with Sertoli cells of the corresponding stages. Germ cells and sertoli cells at each differentiation stage were functionally heterogeneous and stage-specific (Figure 8). This suggests that each stage of sperm development requires the assistance of sertoli cells to complete the corresponding stage of sperm development.”

**Reviewer #3 (Recommendations For The Authors):**
The authors revealed 11 germ cell subtypes and 8 Sertoli cell subtypes through single-cell analysis of two OA patients and three NOA patients. And found that the Sertoli cell SC3 subtype (marked by SPATS1) plays an important role in spermatogenesis. It also suggests that Notch1/2/3 signaling and integrins are involved in germ cell-Stotoli cell interactions. This is an interesting and useful article that at least gives us a comprehensive understanding of human spermatogenesis. It provides a powerful tool for further research on NOA. However, there are still some issues and questions that need to be addressed.(1) How to collect testicular tissue, please explain in detail. Extract which part of testicular tissue. It's better to make a schematic diagram.

Thank you for your comments. The process is as follows: Testicular tissues were obtained from two OA patients (OA1-P1 and OA2-P2) and three NOA patients (NOA1-P3, NOA2-P4, NOA3-P5) using micro-dissection of testicular sperm extraction separately.

(2) Whether the tissues of these patients are extracted simultaneously or separately, separated into single cells, and stored, and then single cell analysis is performed simultaneously. Please be specific.

Thank you for your comments. The testicular tissues of these patients were extracted separately, then separated into single cells, and single cell analysis was performed simultaneously.

(3) When performing single-cell analysis, cells from two OA patients were analyzed individually or combined. The same problem occurred in the cells of three NOA patients.

Thank you for your comments. Cells from two OA patients and three NOA patients were analyzed individually.

(4) Can you specifically point out the histological differences between OA and NOA in Figure 1A? This makes it easier for readers to understand the structure change between OA and NOA. Please also label representative supporting cells.

Thank you for your comments. We have revised the description and it was shown in the revised manuscript.

(5) The authors demonstrate that "We speculate that this lack of differentiation may be due to the intense morphological changes occurring in the sperm cells during this period, resulting in relatively minor differences in gene expression." Please provide some verification of this hypothesis? For example, use immunofluorescence staining to observe morphological changes in sperm cells.

Thank you for your comments. Due to limited funds, we will verified this hypothesis in future studies.

(6) The authors demonstrate that " As shown in Figure 5E, we discovered that PRAP1, BST2, and CCDC62 were co-expressed with SOX9 in testes tissues." The staining in Figure 5D is unclear, and it is difficult to explain that SOX9 is co-expressed with PRAP1 BST2 CCDC62 based on the current staining results. The staining patterns of SOX9 (green) and SOX9 (red) are also different. (SOX9 (red) appears as dots, while the background for SOX9 (green) is too dark to tell whether its staining is also in the form of dots.) In summary, increasing the clarity of the staining makes it more convincing. Alternatively, use high magnification to display these results.

Thank you for your comments. I have redyed and updated this part of the immunofluorescence staining results. Please refer to the files named Figure 1, Figure 2, Figure 5, and Figure 8.

(7) In Figure 8, the author emphasized the co-localization of Sertoli cells and Germ cells at corresponding stages and did a lot of staining, but it was difficult to distinguish the specific locations of co-localization, which was similar to Figure 5E. If possible, please mark specific colocalizations with arrows or use high magnification to display these results, in order to facilitate readers to better understand.

Thank you for your comments. We have re-stained and updated this part of the data. Please refer to the immunofluorescence staining data in the updated Figure 8.

(8) The authors emphasize that macrophages may play an important role in spermatogenesis. Therefore, adding relevant macrophage staining to observe the differences in macrophage expression between NOA and OA should better support this idea.

Thank you for your comments. Macrophage-related experiments will be further explored in the future.

(9) Notch1/2/3 signaling and integrin were discovered to be involved in germ cell-Sertoli cell interaction. However there are currently no concrete experiments to support this hypothesis. At least simple verification experiments are needed.

Thank you for your comments. Due to limited funding, studies will be carried out in the future.

(10) Data availability statements should not be limited to the corresponding author, especially for big data analysis. This is crucial to the credibility of this data (Have the scRNA-seq and scATAC-seq in this study been deposited in GEO or other databases, and when will they be released to the public?) The data for such big data analysis needs to be saved in GEO or other databases in advance so that more research can use it.

Thank you for your comments. We have deposited scRNA-seq and scATAC-seq data in NCBI. “ScRNA-seq data have been deposited in the NCBI Gene Expression Omnibus with the accession number GSE202647, and scATAC-seq data have been deposited in the NCBI database with the accession number PRJNA1177103.”